# A Semi-Static Replication Method for Bermudan Swaptions under an Affine Multi-Factor Model

**Jori Hoencamp** [1,*], **Shashi Jain** [2] and **Drona Kandhai** [1]

1 Informatics Institute, University of Amsterdam, Science Park 904, 1098XH Amsterdam, The Netherlands; b.d.kandhai@uva.nl
2 Indian Institute of Science, Department of Management Studies, Bangalore 560012, India; shashijain@iisc.ac.in
* Correspondence: j.h.hoencamp@uva.nl

**Abstract:** We present a semi-static replication algorithm for Bermudan swaptions under an affine, multi-factor term structure model. In contrast to dynamic replication, which needs to be continuously updated as the market moves, a semi-static replication needs to be rebalanced on just a finite number of instances. We show that the exotic derivative can be decomposed into a portfolio of vanilla discount bond options, which mirrors its value as the market moves and can be priced in closed form. This paves the way toward the efficient numerical simulation of xVA, market, and credit risk metrics for which forward valuation is the key ingredient. The static portfolio composition is obtained by regressing the target option's value using an interpretable, artificial neural network. Leveraging the universal approximation power of neural networks, we prove that the replication error can be arbitrarily small for a sufficiently large portfolio. A direct, a lower bound, and an upper bound estimator for the Bermudan swaption price are inferred from the replication algorithm. Additionally, closed-form error margins to the price statistics are determined. We practically study the accuracy and convergence of the method through several numerical experiments. The results indicate that the semi-static replication approaches the LSM benchmark with basis point accuracy and provides tight, efficient error bounds. For in-model simulations, the semi-static replication outperforms a traditional dynamic hedge.

**Keywords:** semi-static replication; Bermudan swaptions; affine term structure models

## 1. Introduction

The financial crisis of 2007–2008 firmly emphasized the importance of quantifying counterparty credit risk (CCR), which is the risk that the counterparty will default on the obligation and fail to fulfill its contractual agreements. Important indicators used to measure and price CCR include expected exposure (EE), potential future exposure (PFE), and various valuation adjustments (xVAs), which reflect credit, funding, and capital costs related to OTC derivative trading Gregory (2015). Most of these metrics depend on the distribution of the potential future losses resulting from a credit event. Due to the complex nature of these distributions, practitioners resort to numerical methods like Monte Carlo (MC) simulation to approximate the quantities. Typically, this involves scenario generation for the underlying risk factors and subsequent valuation of the contract for each time-step on each path Zhu and Pykhtin (2007). The latter is generally considered the most involved aspect because it needs to be carried out for full portfolios. This poses a major computational challenge to financial institutions. Efficient numerical methods for derivative valuation, both on spot and future simulation dates, are therefore highly relevant.

To address this problem, we extend the concept of (semi-)static replication, which has been extensively studied for, for example, equity derivatives, to interest rate derivatives. A traditional dynamic replication, such as a delta hedge, is achieved by constructing an asset portfolio that is rebalanced continuously through time as the market moves. A static replication on the other hand is an asset portfolio that mirrors the value of the

derivative without the need for rebalancing. The weights of the portfolio composition are so to speak static. In this work, we consider a semi-static hedge, which is a replicating portfolio that needs to be updated on only a finite number of instances. Considering a replication of vanilla products instead of the exotic derivative itself can greatly simplify its risk-assessment. Typically, ample machinery is available to analyze vanilla instruments, including closed-form prices and sensitivities.

In the equity world, the static replication problem has been addressed in the literature by, for example, Breeden and Litzenberger (1978), Carr and Bowie (1994), Carr et al. (1999), and Carr and Wu (2014). The main concept is to construct an infinite portfolio of short-dated European options with a continuum of different strike prices. A different but comparable approach is proposed in Derman et al. (1995). Here, a portfolio of European options with a continuum of different maturities is constructed to replicate the boundary and terminal conditions of exotic derivatives, such as knock-out options. The replication of an American-style option is challenging as it involves a time-dependent exercise boundary, giving rise to a free boundary problem. In Chung and Shih (2009), this is addressed by composing a portfolio of European options with multiple strikes and maturities, and, in Lokeshwar et al. (2022), a semi-static hedge is constructed using shallow neural network approximations. However, in the field of interest rate (IR) modeling, this topic has received little attention and the static replication of exotic IR derivatives remains largely an open problem. Where equity options depend on the realization of a stock, IR derivatives depend on the realization of a full term structure of interest rates, leveraging the complexity of the hedge. The articles of Pelsser (2003) and Hagan (2005) are among the few contributions to the literature, treating the static replication of guaranteed annuity options, and CMS swaps, caps, and floors, respectively, with a portfolio of European swaptions.

In this work, we study the replication problem of Bermudan swaptions under an affine term structure model, possibly multi-factor. Bermudan swaptions are a class of exotic interest rate derivatives that are heavily traded in the OTC market. We show that such a contract can be semi-statically replicated by a portfolio of short-maturity options, such as discount bond options. We propose a regress-later approach, which is introduced in Lokeshwar et al. (2022) for callable equity options. In Lokeshwar et al. (2022), the replication method combines the approximation power of artificial neural networks (ANNs) with the computational benefits of regress-later schemes. In traditional regress-now schemes, such as that of Longstaff and Schwartz (2001), sampled realizations of the continuation value are regressed against the realizations of the risk factors at the preceding monitor date. Advanced variations in this algorithm, where the polynomial regression functions are replaced by ANNs, include the work of Kohler et al. (2010), Lapeyre and Lelong (2019), and Becker et al. (2020). In contrast, in regress-later schemes, the sampled realizations of the continuation value are regressed against the realizations of the risk factors at *the same* date. The continuation value at the preceding monitor date is then obtained by evaluating the conditional expectation of this regression. An analysis and discussion of the benefits of this approach can be found in Glasserman and Yu (2004) and an example of such a scheme is presented in Jain and Oosterlee (2015).

Novel pricing algorithms that replace costly valuation functions with ANN-based approximations have been the subject of many recent papers. An early attempt to approximate option prices in the Black–Scholes model can be attributed to Hutchinson et al. (1994) and dates back to 1994. Since then, a great number of variations in this approach have been investigated. A comprehensive overview of articles devoted to this topic can be found in the literature review of Ruf and Wang (2020). An accessible introduction to neural networks and an application to derivative valuation is, for example, given in the work of Ferguson and Green (2018). A drawback of directly replacing value functions with ANNs is that the method continues to rely on external pricing methodologies to provide input to the training process. In that sense, it can accelerate, but not fully substitute, traditional valuation routines.

Other approaches in the literature consider an indirect use of ANNs and therefore do not depend on classical benchmarks for training. A noteworthy example is the development of deep backward SDE solvers, which, in a financial context, have been introduced by Henry-Labordere (2017). Where the dynamics of financial risk factors are typically captured by forward SDEs, option prices tend to be the solution to backward SDEs. An application to Bermudan swaption valuation is treated in Wang et al. (2018) and a generalization to a CCR management framework is proposed in Gnoatto et al. (2020). Another example is the development of the deep optimal stopping (DOS) algorithm by Becker et al. (2019). They propose an ANN-based method by directly learning the optimal stopping strategy of callable options, without depending on the approximation of continuation values. In the work of Andersson and Oosterlee (2021), the DOS algorithm is applied to compose exposure profiles for Bermudan contracts.

Our contribution to the existing literature is threefold. First, we propose a semi-static replication method for Bermudan swaptions under a multi-factor short-rate model. In the one-factor case, we argue that replication can be achieved with an options portfolio written on a single discount bond. In the multi-factor case, replication can be achieved with an options portfolio written on a basket of discount bonds. As such, we generalize the Black–Scholes-embedded method presented in Lokeshwar et al. (2022) to an interest rate modeling framework. Additionally we propose an alternative ANN design, such that a replication with vanilla options can also be achieved in the multi-factor case (as opposed to basket options). This facilitates highly efficient pricing, which is essential for credit risk applications, such as exposure, VaR, and xVAs, which rely on frequent re-evaluations of the portfolio.

Second, we propose a direct estimator and a lower and an upper bound estimator to the contract's value, which is implied by the semi-static replication. The lower bound results from applying a non-optimal exercise strategy on an independent set of Monte Carlo paths. The upper bound is based on the dual formulation of Haugh and Kogan (2004) and Rogers (2002), which, in contrast to other work, can be obtained without resorting to expensive nested simulations. We complement the study of Lokeshwar et al. (2022) by deriving analytical error margins to the lower and upper bound estimators. This provides a direct insight toward the approximation quality of the proposed estimators and proves their convergence as the regression errors of the ANNs diminish.

Thirdly, we prove that any desired level of accuracy can be achieved in the replication due to the universal approximating power of ANNs. We support this theoretical result with a range of representative numerical experiments. We demonstrate the pricing accuracy of the proposed algorithm by benchmarking to the established least-square method of Longstaff and Schwartz (2001). The regression error and convergence of the method is presented for different contract specifications. Lastly, we study the replication performance for different ANN designs.

The paper is organized as follows: Section 2 introduces the mathematical setting, describes the modeling framework, and provides the problem formulation. Section 3 provides a thorough introduction to the algorithm, motivates the use and interpretation of neural networks, and treats the fitting procedure. Section 4 introduces the lower bound and upper bound estimates to the true option price. In Section 5, we introduce the error bounds on the direct, lower bound, and upper bound estimates brought forth by the algorithm. We finalize the paper by illustrating the method through several numerical examples in Section 6 and providing a conclusion in Section 7.

## 2. Mathematical Background

In this section, we describe the general framework for our computations and give a detailed introduction to the Bermudan swaption pricing problem.

### 2.1. Model Formulation

We consider a continuous-time financial market, defined on finite time horizon $[0, \bar{T}]$. We additionally consider a probability space $(\Omega, \mathcal{F}, \mathbb{P})$, which represents all possible states of the economy, and let the filtration $\mathbb{F} = (\mathcal{F}_t)_{t \in [0, \bar{T}]}$ represent all information generated by the economy up to time-$t$. The market is assumed to be frictionless and we ignore any transaction costs.

We let $B(t)$ denote the time$-t$ value of the *bank account*. Investments in the money market are assumed to compound a continuous, risk-free interest $r_t$, which we refer to as the *short rate*. $B(t)$ corresponds to the time-$t$ value of a unit of currency invested in the money market at time-zero and we assume it is given by the following expression (see Andersen and Piterbarg 2010a or Brigo and Mercurio 2006):

$$B(t) := e^{\int_0^t r(u)du}, \qquad t \in [0, \bar{T}]$$

We denote by $\mathbb{Q}$ the risk-neutral measure equivalent to $\mathbb{P}$, which is associated to $B(t)$ as the numéraire. Attainable claims denominated by the numéraire are assumed to be martingales under $\mathbb{Q}$, which guarantees the absence of arbitrage Harrison and Pliska (1981).

We assume that the dynamics of the short-rate $r$ are captured by an affine term structure model, in accordance with the set-up introduced in Duffie and Kan (1996) and Dai and Singleton (2000). The short rate itself is therefore considered to be an affine function of a—possibly multi-dimensional—latent factor $\mathbf{x}_t$, i.e.,

$$r(t) = \omega_1 + \boldsymbol{\omega_2}^\top \mathbf{x}_t \tag{1}$$

with $\omega_1$, $\boldsymbol{\omega_2}$ denoting a scalar and a vector of time-dependent coefficients, respectively. We furthermore assume that the stochastic process $\{\mathbf{x}_t\}_{t \in [0,T]}$ is a bounded Markov process that takes values in $\mathbb{R}^d$, which represents all market influences affecting the state of the short rate. Let the dynamics of $\mathbf{x}_t$ be governed by an SDE of the form

$$d\mathbf{x}_t = \mu(t, \mathbf{x}_t)dt + \sigma(t, \mathbf{x}_t)d\mathbf{W}_t \tag{2}$$

where $\mathbf{W}_t$ denotes an $\mathbb{R}^d-$ valued Brownian motion under $\mathbb{Q}$ adapted to the filtration $\mathbb{F}$. The measurable functions $\mu : [0, T] \times \mathbb{R}^d \to \mathbb{R}^d$ and $\sigma : [0, T] \times \mathbb{R}^d \to \mathbb{R}^{d \times d}$ are taken to satisfy the standard regularity conditions by which the SDE in Equation (2) admits a strong solution.

We let $P(t, T)$ denote the time$-t$ value of a zero-coupon bond contract that matures at $T$. A zero-coupon bond guarantees the holder one unit of currency at maturity, i.e., $P(T, T) := 1$. Within the class of affine term structure models, zero-coupon bond prices are exponential affine in $\mathbf{x}_t$ Andersen and Piterbarg (2010b); Duffie and Kan (1996). Therefore, the value of $P(t, T)$ can be expressed as

$$P(t, T) := \mathbb{E}^{\mathbb{Q}}\left[ e^{-\int_t^T r_u du} \Big| \mathcal{F}_t \right] = \exp\left\{ A(t, T) - B(t, T)^\top x_t \right\}$$

where the deterministic coefficients $A(t, T) \in \mathbb{R}$ and $B(T_1, T_2) \in \mathbb{R}^d$ can be found by solving a system of ODEs, which are of the form of the well-known Riccati equations; see Duffie and Kan (1996) or Filipovic (2009) for details. We consider this framework as it is still intensively used for risk management purposes. High-dimensional models, such as Libor market models, can be intractable for quantifying credit risk for large portfolios, particularly in a multi-currency setting. Multi-factor short-rate models are therefore popular amongst practitioners, providing a solid compromise between modeling flexibility and analytical tractability.

For simplicity, we will assume that the collateral rate used for discounting and the instantaneous rate used to derive term rates are both implied by the same short rate $r_t$.

Thus, we consider a classic single-curve model environment. As term rates, we consider simply compounded rates, which we refer to as LIBOR Brigo and Mercurio (2006)

$$L(t, T) := \frac{1 - P(t, T)}{\tau P(t, T)}$$

where $\tau$ denotes the year fraction between date $t$ and $T$.

### 2.2. The Bermudan Swaption Pricing Problem

We consider the pricing problem of a Bermudan swaption. A Bermudan swaption is a contract that gives the holder the right to enter a swap with fixed maturity at a number of predefined monitor dates. Should the holder at any of the monitor dates decide to exercise the option, the holder immediately enters the underlying swap. The lifetime of this swap is assumed to be equal to the time between the exercise date and a fixed maturity date $T_M$.

As an underlying, we take a standard interest rate swap that exchanges fixed versus floating cashflows. For simplicity, we will assume that the contract is priced in a single-curve framework and that cashflow schemes of both legs coincide, yielding fixing dates $\mathcal{T}_f = \{T_0, \ldots, T_{M-1}\}$ and payment dates $\mathcal{T}_p = \{T_1, \ldots, T_M\}$. However, we stress that the algorithm is applicable to any industry standard contract specifications and is not limited to the simplifying assumptions that are made here. The time fraction between two consecutive dates is denoted as $\Delta T_m = T_m - T_{m-1}$. Let $N$ be the notional and $K$ the fixed rate of the swap. Assuming that the holder of the option exercises at $T_m$, the payments of the swap will occur at $T_{m+1}, \ldots, T_M$.

We consider the class of pricing problems, where the value of the contract is completely determined by the Markov process $\{\mathbf{x}_t\}_{t \in [0,T]}$ in $\mathbb{R}^d$ as defined in Section 2. Let $h_m : \mathbb{R}^d \to \mathbb{R}$ be the $\mathcal{F}_{T_m}$-measurable function denoting the immediate pay-off of the option if exercised at time $T_m$. Although the methodology holds for any generalization of the functions $h_m$, we will consider those in accordance with the contract specifications described above. This means that the functions $h_m$ are assumed to be given by

$$h_m(\mathbf{x}_{T_m}) := \delta \cdot N \cdot A_{m,M}(T_m)(S_{m,M}(T_m) - K)$$

where the indicator $\delta = 1$ infers a payer and $\delta = -1$ infers a receiver swaption. The swap rate $S_{m,M}$ and the annuity $A_{m,M}$ are defined in the same fashion as Brigo and Mercurio (2006), given by the expressions

$$S_{m,M}(t) = \frac{\sum_{j=m+1}^{M} \Delta T_j P(t, T_j) F(t, T_{j-1}, T_j)}{\sum_{j=m+1}^{M} \Delta T_j P(t, T_j)}, \quad A_{m,M}(t) = \sum_{j=m+1}^{M} \Delta T_j P(t, T_j)$$

where the function $F$ denotes the simply compounded forward rate given by the expression

$$F(t, T_{j-1}, T_j) = \frac{1}{\Delta T_j} \left( \frac{P(t, T_{j-1})}{P(t, T_j)} - 1 \right)$$

for any $j \in \{1, \ldots, M\}$. For details, we refer to Brigo and Mercurio (2006).

Now, let $\mathbb{T}$ denote the set of all discrete stopping times with respect to the filtration $\mathbb{F}$, taking values on the grid $\mathcal{T}_f \cup \{\infty\}$. Define the function $h_\tau$ as

$$h_\tau(\mathbf{x}_\tau) := h_{\tau(\omega)}(\mathbf{x}_\tau(\omega)) = \begin{cases} h_m(\mathbf{x}_{T_m}) & \text{if } \tau(\omega) = T_m \\ 0 & \text{if } \tau(\omega) = \infty \end{cases}, \quad \omega \in \Omega \tag{3}$$

In this notation, $\tau(\omega) = \infty$ indicates that the option is not exercised at all. We aim to approximate the time-zero value of the Bermudan swaption, which satisfies the following equation:

$$V(0) = \sup_{\tau \in \mathbb{T}} \mathbb{E}^{\mathbb{Q}} \left[ \frac{h_\tau(\mathbf{x}_\tau)}{B(\tau)} \bigg| \mathcal{F}_0 \right] \tag{4}$$

Finding the optimal exercise strategy $\tau$ is typically a non-trivial exercise. Numerical approximations for $V(0)$ can, however, be computed by considering a dynamical programming formulation as given below, which is shown to be equivalent to (4) in, for example, Glasserman (2013). Let $t \in (T_m, T_{m+1}]$ for some $m \in \{0, \dots, M-2\}$ and denote by $V(t)$ the value of the option, conditioned on the fact that it is not yet exercised prior to $t$. This value satisfies the equation (see Glasserman 2013)

$$V(t) = \begin{cases} \max\{h_{M-1}(\mathbf{x}_{T_{M-1}}),\, 0\} & \text{if } t = T_{M-1} \\ \max\left\{h_m(\mathbf{x}_t),\, B(t)\mathbb{E}^{\mathbb{Q}}\left[\frac{V(T_{m+1})}{B(T_{m+1})}\bigg|\mathcal{F}_t\right]\right\} & \text{if } t = T_m,\, m \in \{0, \dots, M-2\} \\ B(t)\mathbb{E}^{\mathbb{Q}}\left[\frac{V(T_{m+1})}{B(T_{m+1})}\bigg|\mathcal{F}_t\right] & \text{if } t \in (T_m, T_{m+1}),\, m \in \{0, \dots, M-2\} \end{cases} \tag{5}$$

We refer to the random variables $C_m(t) := B(t)\mathbb{E}^{\mathbb{Q}}\left[\frac{V(T_{m+1})}{B(T_{m+1})}\big|\mathcal{F}_t\right]$ as the hold or continuation values. They represent the expected value of the contract if it is not being exercised up until $t$ but continues to follow the optimal policy thereafter. Approximations of the dynamic formulation are typically obtained by a backward iteration based on simulations of the underlying risk factors. The objective is then to determine the continuation values as a function of the state of the risk factor $\mathbf{x}_t$. Popular numerical schemes based on regression have been introduced in, for example, Carriere et al. (1996) and Longstaff and Schwartz (2001).

Based on approximations of the continuation values, the optimal policy $\tau$ can be computed as follows. Assume that, for a given scenario $\omega \in \Omega$, the risk factor takes the values $\mathbf{x}_{T_0} = x_0, \dots, \mathbf{x}_{T_{M-1}} = x_{M-1}$. Then, the holder should continue to hold the option if $C_m(T_m) > h_m(x_m)$ and exercise as soon as $C_m(T_m) \leq h_m(x_m)$. In other words, the exercise strategy can be determined as

$$\tau(\omega) = \min\left\{ T_m \in \mathcal{T}_f \big| C_m(T_m) \leq h_m(x_m) \right\}$$

Should, for some scenario, the continuation value be bigger than the immediate pay-off for each monitor date, then $\tau(\omega) = \infty$ and the option expires as worthless.

## 3. A Semi-Static Replication for Bermudan Swaptions

The main concept of our method is to construct static hedge portfolios that replicate the dynamical formulation in Equation (5) between two consecutive monitor dates. In this section, we introduce the algorithm for a Bermudan swaption that is priced under a multi-factor affine term structure model. The methodology is inspired by the algorithm presented in Lokeshwar et al. (2022) and utilizes a regress-later technique in which the intermediate option values are regressed against simple IR assets, such as discount bonds. The regression model is chosen deliberately to represent the pay-off of an options portfolio written on these assets. An important consequence is that the hedge can be valued in closed form. Throughout this work, we will use the terms semi-static hedge and semi-static replication interchangeably. A hedge in general refers to a trading strategy that reduces the exposure to market risk of an outstanding position. A replication refers to an asset portfolio that mirrors the value of a derivative, which is a common means to set up a hedge. As we see the efficient valuation properties in the context of credit risk quantification as the main application, rather than actual hedging, we will put emphasis on the term replication.

### 3.1. The Algorithm

The regress-later algorithm is executed in an iterative manner, backward in time. The outcome is a set of option portfolios $\{\Pi_{M-1}, \ldots, \Pi_0\}$ written on pre-selected IR assets. To be more precise, the algorithm determines the weights and strikes of each portfolio $\Pi_m$, such that it closely mirrors the Bermudan swaption after its composition at $T_{m-1}$ until its expiry at $T_m$. The pay-off of $\Pi_m$ exactly meets the cost of composing the next portfolio $\Pi_{m+1}$ or the Bermudan's pay-off in case it is exercised. The methodology yields a semi-static hedging strategy as the portfolio compositions are constant between two consecutive monitor dates. Hence, there is no need for continuous rebalancing, as is the case for a dynamic hedging strategy. The algorithm can roughly be divided into three steps, presented below. Algorithm 1 summarizes the method.

---

**Algorithm 1** The algorithm for a Bermudan swaption

---

Generate $N$ risk factor scenarios for $\mathbf{x}_{T_m}$ for $m = 0, \ldots, M$
Compute $N$ corresponding asset scenarios for $z_m$ for $m = 0, \ldots, M$
$\tilde{V}\left(T_{M-1}; x^n_{T_{M-1}}\right) \leftarrow \max\left\{h_{M-1}\left(x^n_{T_{M-1}}\right), 0\right\}$ for $n = 1, \ldots, N$
Initialize $G_{M-1}$ parameters $\xi_{M-1}$ from independent uniform distributions
**for** $m = M-1, \ldots, 1$ **do**
    $\xi_m \leftarrow \underset{\xi \in \mathbb{R}^p}{\text{argmin}}\ L(\xi | \hat{z}_m, \hat{x}_m)$ minimizing the MSE
    **for** $n = 1, \ldots, N$ **do**
        $\tilde{C}_{m-1}(T_{m-1}) \leftarrow B(T_{m-1})\mathbb{E}^{\mathbb{Q}}\left[\frac{G_m(z_m(T_m))}{B(T_m)} \Big| \mathcal{F}_{T_{m-1}}\right]$
        $\tilde{V}(T_{m-1}; x^n_{T_{m-1}}) \leftarrow \max\left\{\tilde{C}_{m-1}(T_{m-1}), h_{m-1}\left(x^n_{T_{m-1}}\right)\right\}$
    **end for**
    $\xi_{m-1} \leftarrow \xi_m$ initialize weights of $G_{m-1}$
**end for**
$\xi_0 \leftarrow \underset{\xi \in \mathbb{R}^p}{\text{argmin}}\ L(\xi | \hat{z}_0, \hat{x}_0)$ minimizing the MSE
**return** $\mathbb{E}^{\mathbb{Q}}\left[\frac{G_0(z_0(T_0))}{B(T_0)} \Big| \mathcal{F}_0\right]$

---

#### 3.1.1. Sample the Independent Variables

We start by sampling $N$ realizations of the risk factor $\mathbf{x}_t$ on the time grid $\mathcal{T} = \{T_0, \ldots, T_{M-1}\}$. These realizations will serve as an input for the regression data. We will denote the data points as $\hat{x} := \left\{\left(x^n_{T_0}, \ldots, x^n_{T_{M-1}}\right)\right\}_{n=1}^N$. Different sample methodologies could be used, such as:

- Take a standard quadrature grid for each monitor date $T_m$, associated with the transition density of the risk factor. For example, if $\mathbf{x}_t$ has Gaussian dynamics, one could consider the Gauss–Hermite quadrature scaled and shifted in accordance with the mean and variance of $\mathbf{x}_t$. See, for example, Xiu (2010).
- Discretize the SDE of the risk factor and sample by the means of an Euler or Milstein scheme. Make sure that a sufficiently coarse time-stepping grid is used, which includes the $M$ monitor dates. See, for example, Kloeden and Platen (2013) for details.

Secondly, we select an asset that will serve as the independent variable for the regression. We will denote this asset as $z_m(t)$. The choice for $z_m$ can be arbitrary, as long as it meets the following conditions:

- The asset $z_m(T_m)$ should be a square integrable random variable that is $\mathcal{F}_{T_m}$ measurable, taking values in $\mathbb{R}^d$.
- The risk-neutral price of $z_m(t)$ should only be dependent on the current state of the risk factor and be almost surely unique; that is, the mapping $\mathbf{x}_{T_m} \mapsto z_m | \mathbf{x}_{T_m}$ should be

continuous and injective. This is required to guarantee a well-defined parametrization of the option value.

Examples for $z_m$ would be a zero-coupon bond, a forward Libor rate, or a forward swap rate. For each sampled realization of the risk factor, the corresponding realization of the asset value will be computed and denoted as $\hat{z} := \left\{ \left( z_0^n, \ldots, z_{M-1}^n \right) \right\}_{n=1}^N$. This will serve as the regression data in the following step.

### 3.1.2. Regress the Option Value against an IR Asset

In this phase, we compose replication portfolios $\Pi_0, \ldots, \Pi_{M-1}$ by fitting $M$ regression functions $G_0, \ldots, G_{M-1}$. We consider functions of the form $G_m : \mathbb{R}^d \to \mathbb{R}$, which assign values in $\mathbb{R}$ to each realization of the selected asset $z_m$. Fitting is performed recursively, starting at $T_{M-1}$, moving backwards in time, until the first exercise opportunity $T_0$. Approximations of the Bermudan swaption value at each monitor date serve as the dependent variable. At the final monitor date, the value of the contract (given it has not been exercised) is known to be

$$V\left(T_{M-1}; x_{T_{M-1}}^n\right) = \max\left\{ h_{M-1}\left(x_{T_{M-1}}^n\right), 0 \right\}, \qquad n = 1, \ldots, N$$

Now, assume that, for some monitor date $T_m$, we have an approximation of the contract value $\tilde{V}\left(T_m; x_{T_m}^n\right) \approx V\left(T_m; x_{T_m}^n\right)$. Let $\xi_m \in \mathbb{R}^p$ for some $p \in \mathbb{N}$ denote the vector of the unknown regression parameters. The objective is to determine $\xi_m$ such that

$$G_m(z_m(T_m)) \approx V(T_m)$$

with the smallest possible error. This is carried out by formulating and solving a related optimization problem. In this case, we choose to minimize the expected square error, given by

$$\mathbb{E}^{\mathbb{Q}}\left[ \left| G_m(z_m(T_m)) - V(T_m) \right|^2 \right] \tag{6}$$

There is no exact analytical expression available for the expectation of Equation (6). However, it can be approximated using the sampled regression data, giving rise to an empirical loss function $L$ given by

$$L(\xi_m | \hat{z}_m, \hat{x}_m) = \frac{1}{N} \sum_{n=1}^N \left( G_m(z_m^n) - \tilde{V}\left(T_m; x_{T_m}^n\right) \right)^2 \tag{7}$$

The parameters $\xi_m$ are then the result of the fitting procedure, such that

$$\xi_m \approx \underset{\xi \in \mathbb{R}^p}{\operatorname{argmin}} \ L(\xi | \hat{z}_m, \hat{x}_m)$$

If the regression model is chosen accordingly, $G_m(z_m)$ represents the pay-off at $T_m$ of a derivative portfolio $\Pi_m$ written on the selected asset $z_m$. Details on suggested functional forms of $G_m$, asset selection for $z_m$, and fitting procedures are subject of Section 3.2.

### 3.1.3. Compute the Continuation Value

Once the regression is completed, the last step is to compute the continuation value and subsequently the option value at the monitor date preceding $T_m$. For each scenario $n = 1, \ldots, N$, we approximate the continuation value as

$$\begin{aligned}
\tilde{C}_{m-1}(T_{m-1}) &= B(T_{m-1})\mathbb{E}^{\mathbb{Q}}\left[ \frac{\tilde{V}(T_m)}{B(T_m)} \middle| \mathcal{F}_{T_{m-1}} \right] \\
&\approx B(T_{m-1})\mathbb{E}^{\mathbb{Q}}\left[ \frac{G_m(z_m(T_m))}{B(T_m)} \middle| \mathcal{F}_{T_{m-1}} \right]
\end{aligned} \tag{8}$$

As $G_m$ is chosen to represent the pay-off of a derivative portfolio $\Pi_m$ written on $z_m$, we argue that computing $C_{m-1}$ is in fact equivalent to the risk-neutral pricing of $\Pi_m$. In other words, we have

$$\tilde{C}_{m-1}(T_{m-1}) = B(T_{m-1})\mathbb{E}^{\mathbb{Q}}\left[\frac{\Pi_m(T_m)}{B(T_m)}\middle|\mathcal{F}_{T_{m-1}}\right] := \Pi_m(T_{m-1})$$

In Section 3.2, we treat examples for which $\Pi_m$ can be computed in closed form.

Finally, the option value at the preceding monitor date $T_{m-1}$ is given by

$$\tilde{V}\left(T_m; x_{T_m}^n\right) = \max\left\{\tilde{C}_{m-1}(T_{m-1}), \, h_{m-1}\left(x_{T_{m-1}}^n\right)\right\}, \qquad n = 1, \dots, N$$

The steps are repeated recursively until we have a representation $G_0$ of the option value at the first monitor date. An estimator of the time-zero option value is given by

$$\tilde{V}(0) = \mathbb{E}^{\mathbb{Q}}\left[\frac{G_0(z_0(T_0))}{B(T_0)}\middle|\mathcal{F}_0\right]$$

We refer to this approximation as the *direct estimator*.

### 3.2. A Neural Network Approach to $G_m$

In this section, we propose to represent the regression functions $G_m$ as shallow, artificial neural networks. The choices that are presented here are adapted to a framework of Gaussian risk factors, such as that presented in Section 2. The method, however, lends itself to be generalized to a broader class of models by considering an appropriate adjustment to the input or structure.

#### 3.2.1. The 1-Factor Case

First, we discuss the case $d = 1$. Let $m \in \{0, \dots, M-1\}$. As a regression function, we consider a fully connected, feed-forward neural network with one hidden layer, denoted as $G_m : \mathbb{R} \to \mathbb{R}$. The design with only a single hidden layer is graphically represented in Figure 1 and is chosen deliberately to facilitate the network's interpretation. As an input to the network (the asset $z_m$), we select a zero-coupon bond, which pays one unit of currency at $T_M$.

- The first layer consists of a single node and corresponds to the discount bond price, which serves as input. It is represented by the left node in Figure 1. The hidden layer has $q \in \mathbb{N}$ hidden nodes, represented by the center layer in Figure 1. The affine transformation acting between the first two layers is denoted $A_1 : \mathbb{R} \to \mathbb{R}^q$ and is of the form

$$A_1 : x \mapsto \mathbf{w}_1 x + \mathbf{b}, \qquad \mathbf{w}_1 \in \mathbb{R}^{q \times 1}, \mathbf{b} \in \mathbb{R}^q$$

  As an activation function $\varphi : \mathbb{R}^q \to \mathbb{R}^q$ acting on the hidden layer, we take the ReLU-function, given by

$$\varphi : (x_1, \dots, x_q) \mapsto (\max\{x_1, 0\}, \dots, \max\{x_q, 0\})$$

  Note that the ReLU function corresponds to the pay-off function of a European option.

- The output of the network estimates contract value $\tilde{V}_m \in \mathbb{R}$ and therefore takes value in $\mathbb{R}$. It is represented by the right node in Figure 1. We consider a linear transformation acting between the second and last layer $A_2 : \mathbb{R}^q \to \mathbb{R}$, given by

$$A_2 : x \mapsto \mathbf{w}_2 x, \qquad \mathbf{w}_2 \in \mathbb{R}^{1 \times q}$$

  On top of that, we apply the linear activation, which comes down to an identity function, mapping $x$ to itself.

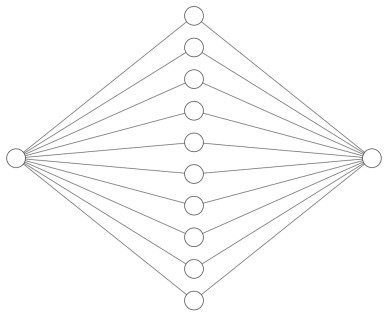

**Figure 1.** Suggested neural network design for $Dim(\mathbf{x}_t) = 1$.

Combined together, the network is specified to satisfy

$$G_m(\cdot) := A_2 \circ \varphi \circ A_1$$

and the trainable parameters can be presented by the list

$$\xi_m = \big\{ w_{1,1}, b_{1,1}, \dots, w_{1,q}, b_{1,q} \big\} \cup \big\{ w_{2,1}, \dots, w_{2,q} \big\}$$

3.2.2. Interpretation of the Neural Network

Now that we have specified the structure of the neural network, we will discuss how each function $G_m$ can be interpreted as a portfolio $\Pi_m$. In the one-dimensional case, $G_m$ can be expressed as follows:

$$G_m(z_m) := \sum_{j=1}^{q} w_{2,j} \max\{w_{1,j} z_m + b_j, \, 0\}$$

We can regard this as the pay-off of a derivative portfolio $\Pi_m$ written on the asset $z_m$. The portfolio contains $q$ derivatives that each have a terminal value equal to $w_{2,j}$ $\max\{w_{1,j} z_m + b_j, \, 0\}$. In total, we can recognize four types of products, which depend on the signs of $w_{1,j}$ and $b_j$.

1. If $w_{1,j} > 0$ and $b_j > 0$, we have

$$w_{2,j} \max\{w_{1,j} z_m + b_j, \, 0\} = w_{2,j} w_{1,j} z_m + w_{2,j} b_j$$

which is the pay-off of a forward contract on $w_{2,j} w_{1,j}$ units in $z_m$ and $w_{2,j} b_j$ units of currency.

2. If $w_{1,j} > 0$ and $b_j < 0$, we have

$$w_{2,j} \max\{w_{1,j} z_m + b_j, \, 0\} = w_{2,j} w_{1,j} \max\left\{ z_m - \frac{-b_j}{w_{1,j}}, \, 0 \right\}$$

which is the pay-off corresponding to $w_{2,j} w_{1,j}$ units of a European call option written on $z_m$, with strike price $\frac{-b_j}{w_{1,j}}$.

3. If $w_{1,j} < 0$ and $b_j > 0$, we have

$$w_{2,j} \max\{w_{1,j} z_m + b_j, \, 0\} = -w_{2,j} w_{1,j} \max\left\{ \frac{b_j}{-w_{1,j}} - z_m, \, 0 \right\}$$

which is the pay-off corresponding to $-w_{2,j} w_{1,j}$ units of a European put option written on $z_m$, with strike price $\frac{b_j}{-w_{1,j}}$.

4.    If $w_{1,j} < 0$ and $b_j < 0$, we have

$$w_{2,j} \max\{w_{1,j} z_m + b_j, \, 0\} = 0$$

which clearly represents a worthless contract.

The sign of the coefficient $w_{2,j}$ indicates if one has a short or long position of the product in the portfolio. Hence, under the assumption of a frictionless economy, the absence of arbitrage, and the Markov property for $z_m$, the portfolio $\Pi_m$ replicates the original Bermudan contract over the period $(T_{m-1}, T_m]$. As the portfolio composition is constant between two consecutive monitor dates, the method described here can be interpreted as a semi-static hedging strategy.

### 3.2.3. The Multi-Factor Case

In the case $d \geq 2$, we propose that a basket of $d$ zero-coupon bonds all maturing at different dates $T_m + \delta_1, \ldots, T_m + \delta_n$ is required as input to the regression. If the risk factor space is $d$-dimensional, it can only be parametrized by an at least $d$-dimensional asset vector.

To see why the above statement is true, simply consider $n$ bonds $P(T_m, T_m + \delta_1), \ldots,$ $P(T_m, T_m + \delta_n)$ and note that the following relation holds:

$$\begin{pmatrix} P(T_m, T_m + \delta_1) \\ \vdots \\ P(T_m, T_m + \delta_n) \end{pmatrix} = \begin{pmatrix} \exp\{A(T_m, T_m + \delta_1) - \sum_{j=1}^d B_j(T_m, T_m + \delta_1) x_j(T_m)\} \\ \vdots \\ \exp\{A(T_m, T_m + \delta_n) - \sum_{j=1}^d B_j(T_m, T_m + \delta_n) x_j(T_m)\} \end{pmatrix}$$

$$\implies \begin{pmatrix} B_1(T_m, T_m + \delta_1) & \ldots & B_d(T_m, T_m + \delta_1) \\ \vdots & \ddots & \vdots \\ B_1(T_m, T_m + \delta_n) & \ldots & B_d(T_m, T_m + \delta_n) \end{pmatrix} \begin{pmatrix} x_1(T_m) \\ \vdots \\ x_d(T_m) \end{pmatrix}$$

$$= \begin{pmatrix} A(T_m, T_m + \delta_1) - \log P(T_m, T_m + \delta_1) \\ \vdots \\ A(T_m, T_m + \delta_d) - \log P(T_m, T_m + \delta_n) \end{pmatrix}$$

$$\implies \mathbf{B}(T_m) \mathbf{x}_{T_m} = \boldsymbol{\alpha}$$

Since we have that $rank(\mathbf{B}(T_m)) = \min\{n, d\}$, it follows that if $n < d$, the image of $\mathbf{B}$ does not span the whole risk factor space, whereas if $n > d$, the image of $\mathbf{B}$ is still equal to the case $n = d$.

Concluding on the argument above, it would be an obvious choice to take a $d-$dimensional vector of bonds as the input and generalize the architecture of $G_m$ by increasing the input dimension (i.e., the number of nodes in the first layer) from 1 to $d$. However, in that case, $\Pi_m$ represents a derivatives portfolio written on a basket of bonds, by which the tractability of pricing $\Pi_m$ would be lost. Therefore, we suggest two alternatives to the design of $G_m$, intended to preserve the analytical valuation potential of $\Pi_m$.

The basic specifications of the neural network will remain similar to the one-factor case. We consider a feed-forward neural network with one hidden layer of the form $G_m : \mathbb{R}^d \to \mathbb{R}$.

- The first layer consists of $d$ nodes and the hidden layer has $q \in \mathbb{N}$ hidden nodes. The affine transformation and activation acting between the first two layers are denoted $A_1 : \mathbb{R}^d \to \mathbb{R}^q$ and $\varphi : \mathbb{R}^q \to \mathbb{R}^q$, respectively, given by

$$A_1 : x \mapsto \mathbf{w}_1 x + \mathbf{b}, \qquad \mathbf{w}_1 \in \mathbb{R}^{q \times d}, \mathbf{b} \in \mathbb{R}^q$$
$$\varphi : (x_1, \ldots, x_q) \mapsto (\max\{x_1, 0\}, \ldots, \max\{x_q, 0\})$$

- The output contains a single node. A linear transformation acts between the second and last layer $A_2 : \mathbb{R}^q \to \mathbb{R}$, together with the linear activation, given by

$$A_2 : x \mapsto \mathbf{w}_2 x, \qquad \mathbf{w}_2 \in \mathbb{R}^{1 \times q}$$

- The network is given by $G_m(\cdot) := A_2 \circ \varphi \circ A_1$.

### 3.2.4. Suggestion 1: A Locally Connected Neural Network

The outcome of each node in the hidden layer represents the terminal value of a derivative written on the asset $\mathbf{z}_m$, which, together, compose the portfolio $\Pi_m$. In the $d-$dimensional case, the outcome of the $j^{th}$ node $\nu_j$ can be expressed as

$$\nu_j(\mathbf{z}) = \max\left\{ \sum_{k=1}^{d} w_{jk} z_k + b_j, \ 0 \right\}$$

which corresponds to the pay-off of an arithmetic basket option with weights $w_{j1}, \dots w_{jd}$ and strike price $b_j$. Such an exotic option is difficult to price. To overcome this issue, we constrain the matrix $\mathbf{w}_1$ to only admit a single non-zero value in each row. The architecture of this suggestion is graphically depicted in Figure 2a. Let the number of hidden nodes be a multiple of the input dimension, i.e., $q = n \cdot d$ for some $n \in \mathbb{N}$. The matrix $\mathbf{w}_1$ is set to be of the form

$$\mathbf{w}_1 = \begin{pmatrix} w_{1,1} & 0 & 0 & \cdots & 0 & 0 \\ \vdots & \vdots & \vdots & & \vdots & \vdots \\ w_{1,n} & 0 & 0 & \cdots & 0 & 0 \\ 0 & w_{2,n+1} & 0 & \cdots & 0 & 0 \\ \vdots & \vdots & \vdots & & \vdots & \vdots \\ 0 & w_{2,2n} & 0 & \cdots & 0 & 0 \\ \vdots & \vdots & \vdots & & \vdots & \vdots \\ 0 & 0 & 0 & \cdots & 0 & w_{d,d \cdot n} \end{pmatrix}$$

As a result, none of the hidden nodes are connected to more than one input node (see Figure 2a). Therefore, the outcome of each node $\nu_j$ again represents a European option or forward written on a single bond, which can be priced in closed form (see Appendix A.1).

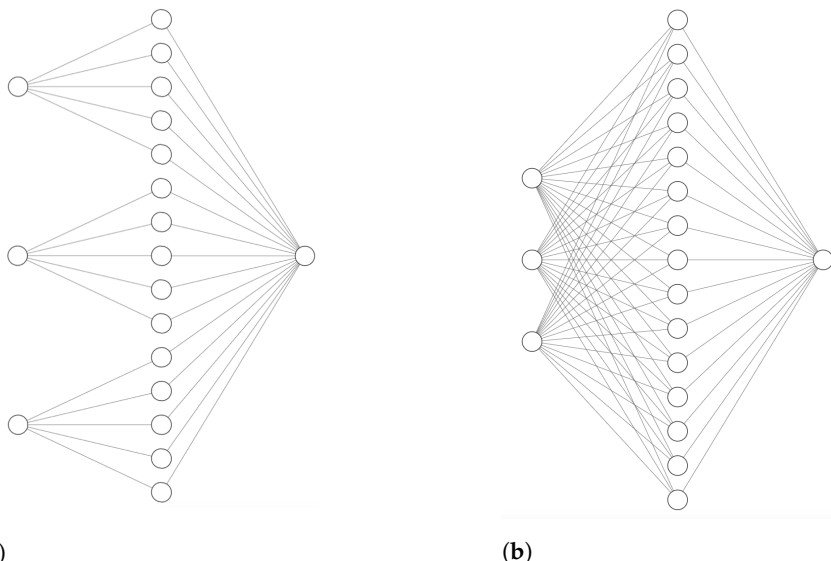

(**a**)            (**b**)

**Figure 2.** Suggested neural network designs for $Dim(\mathbf{x}_t) \geq 2$. (**a**) Locally connected neural network. (**b**) Fully connected neural network.

We can recognize two drawbacks to this approach. First, the number of trainable parameters for a fixed number of hidden nodes is much lower compared to the fully connected case. This can simply be overcome by increasing $q$. Second, as the network is not fully connected, the universal approximation theorem no longer applies to $G_m$. Therefore, we have no guarantee that the approximation errors can be reduced to any desirable level. Our numerical experiments however indicate that the approximation accuracy of this design is not inferior to that of a fully connected counterpart of the same dimensions; see Section 6.

### 3.2.5. Suggestion 2: A Fully Connected Neural Network

Our second approach does not entail altering the structure or weights of the network, but suggests to take a different input. We hence consider a fully connected feed-forward neural network with one hidden layer of the form $G_m : \mathbb{R}^d \to \mathbb{R}$. The architecture is graphically depicted in Figure 2. As a consequence, each hidden node is connected to each input node. However, as an input, we use the log of $n$ bonds, i.e.,

$$\mathbf{z}_m := (\log P(T_m, T_m + \delta_1), \ldots, \log P(T_m, T_m + \delta_n))^\top$$

Therefore, each node $\nu_j$ can be compared to the pay-off of a geometric basket option written on $n$ assets $\mathbf{z}_m$ equal to the log of $P(t, T_m + \delta_j)$. Under the assumption that the dynamics of the risk factor $\mathbf{x}_t$ are Gaussian, these options can be priced explicitly as we will show in Appendix A.2.

An advantage of this approach is that it employs a fully connected network that, by virtue of the universal approximation theorem Hornik et al. (1989), can yield any desired level of accuracy. A drawback is that the financial interpretation of the network as a replicating portfolio is not as strong as in suggestion 1 due to the required log in the payoff.

### 3.3. Training of the Neural Networks

In this section, we specify some of the main considerations related to the fitting procedure of the algorithm. The method requires the training of $M$ shallow feed-forward networks as specified in Section 3.2, which we denote $G_0, \ldots, G_{M-1}$. Our numerical experiments indicated that the normalization of the training set strongly improved the networks' fitting accuracy. Details for pre-processing the regression data are treated in Appendix B.

#### Optimization

The training of each network is performed in an iterative process, starting with $G_{M-1}$ working backwards until $G_0$. The effectiveness of the process depends on several standard choices related to neural network optimization, of which some are listed below.

- As an optimizer, we apply AdaMax Kingma and Ba (2014), a variation of the commonly used Adam algorithm. This is a stochastic, first-order, gradient-based optimizer that updates weights inversely proportional to the $L_\infty$-norm of their current and past gradient, whereas Adam is based on the $L_2$-norm. Our experiments indicate that AdaMax slightly outperforms comparable algorithms in the scope of our objectives.
- The batch size, i.e., the number of training points used per weight update, is set to a standard 32. The learning rate, which scales the step size of each update, is kept in the range 0.0001–0.0005.
- For the initial network, $G_{M-1}$, we use random initialization of the parameters. If the considered contract is a payer Bermudan swaption, we initialize the (non-zero) entries of $\mathbf{w}_1$ i.i.d. unif$(0,1)$ and the biases $\mathbf{b}$ i.i.d. unif$(-1,0)$. In the case of a receiver contract, it is the other way around. The weights $\mathbf{w}_2$ are initialized i.i.d. unif$(-1,1)$.
- For the subsequent networks, $G_{M-2}, \ldots, G_0$, each network $G_m$ is initialized with the final set of weights of the previous network $G_{m+1}$.
- As a training set for the optimizer, we use a collection of 20,000 data-points.

Some specific choices for the hyperparameters are motivated by a convergence analysis presented in Appendix C.

## 4. Lower and Upper Bound Estimates

The algorithm described in Section 3.1 gives rise to a direct estimator of the true option price $V$. The accuracy of this estimator depends on the approximation performance of the neural networks at each monitor date. Should each regression yield a perfect fit, then the estimation error would automatically be zero. In practice, however, the loss function, defined in Equation (7), never fully converges to zero. As the networks are trained to closed-form exercise and continuation values, error measures such as MSE and MAE can be easily obtained. In particular, the mean absolute errors provide a strong indication of the error bounds on the direct estimator (see Section 5).

Although convergence errors put solid bounds on the accuracy of the estimator, they are typically quite loose. Therefore, they give rise to non-tight confidence bounds. To overcome this issue, we introduce a numerical approximation to a tight lower and upper bound to the true price, in the same spirit as Lokeshwar et al. (2022). These should provide a better indication of the quality of the estimate.

### 4.1. The Lower Bound

We compute a lower bound approximation by considering the non-optimal exercise strategy $\tilde{\tau}$ implied by the continuation values estimates introduced in Section 3.1. We define $\tilde{\tau}$ as

$$\tilde{\tau}(\omega) = \min\left\{ T_m \in \mathcal{T}_f \big| \tilde{C}_m(T_m) \leq h_m(\mathbf{x}_{T_m}) \right\} \tag{9}$$

where $\tilde{C}_m$ refers to the approximated continuation value given in Equation (8). A strict lower bound is now given by

$$L(0) = \mathbb{E}^{\mathbb{Q}}\left[ \frac{h_{\tilde{\tau}}(\mathbf{x}_{\tilde{\tau}})}{B(\tilde{\tau})} \bigg| \mathcal{F}_0 \right] = P(0, T_M) \mathbb{E}^{T_M}\left[ \frac{h_{\tilde{\tau}}(\mathbf{x}_{\tilde{\tau}})}{P(\tilde{\tau}, T_M)} \bigg| \mathcal{F}_0 \right] \tag{10}$$

where $h_{\tilde{\tau}}$ corresponds to the definition given in Equation (3). The term on the right is obtained by changing the measure from $\mathbb{Q}$ to the $T_M-$forward measure $\mathbb{Q}^{T_M}$ Geman et al. (1995). Under the $T_M-$forward measure, the lower bound can be estimated by simulating a fresh set of scenarios of the risk factor $\hat{x} := \left\{ \left( x_{t_1}^n, x_{t_2}^n, \ldots, x_{T_M}^n \right) \big| n = 1, \ldots, N \right\}$. Denote by $P^n(t, T_M)$ the zero-coupon bond realization corresponding to $x_t^n$. Then, the lower bound cab be approximated as

$$\tilde{L}(0) = \frac{P(0, T_M)}{N} \sum_{n=1}^{N} \frac{h_{\tilde{\tau}^n}(x_{\tilde{\tau}^n}^n)}{P^n(\tilde{\tau}^n, T_M)}$$

### 4.2. The Upper Bound

We compute an upper bound by considering a dual formulation of the price expression Equation (4) as proposed in Haugh and Kogan (2004) and Rogers (2002). Let $\mathcal{M}$ denote the set of all martingales $M_t$ adapted to $\mathbb{F}$ such that $\sup_{t \in [0,T]} |M_t| < \infty$. An upper bound $U(0)$ to the true price $V(0)$ is obtained by observing that the following inequality holds (see Haugh and Kogan 2004):

$$V(0) \leq M_0 + \mathbb{E}^{\mathbb{Q}}\left[ \max_{T_m \in \mathcal{T}_f} \left\{ \frac{h_m(\mathbf{x}_{T_m})}{B(T_m)} - M_{T_m} \right\} \bigg| \mathcal{F}_0 \right] := U(0) \tag{11}$$

for any $M_t \in \mathcal{M}$. To find a suitable martingale that yields a tight bound, we consider the Doob–Meyer decomposition of the true discounted option price process $\frac{V(t)}{B(t)}$. As the price process is a supermartingale, we can write

$$\frac{V(t)}{B(t)} := Y_t + Z_t$$

where $Y_t$ denotes a martingale and $Z_t$ is a predictable, strictly decreasing process such that $Z_0 = 0$. Note that Equation (11) attains an equality if we set $M_t = Y_t$, i.e., the martingale part of the option price process. The bound will hence be tight if we consider a martingale $M_t$ that is close to the unknown $Y_t$. Let $G_m(\cdot)$ denote the neural networks induced by the algorithm. In the spirit of Andersen and Broadie (2004) and Lokeshwar et al. (2022), we construct a martingale on the discrete time grid $\{0, T_0, \ldots, T_{M-1}\}$ as follows:

$$M_0 = \mathbb{E}^{\mathbb{Q}}\left[\frac{G_0(z_0(T_0))}{B(T_0)}\bigg|\mathcal{F}_0\right], \ M_{T_0} = \frac{G_0(z_0(T_0))}{B(T_0)}$$

$$M_{T_m} = M_{T_{m-1}} + \frac{G_m(z_m(T_m))}{B(T_m)} - \mathbb{E}^{\mathbb{Q}}\left[\frac{G_m(z_m(T_m))}{B(T_m)}\bigg|\mathcal{F}_{T_{m-1}}\right], \quad m = 1, \ldots, M-1 \tag{12}$$

Clearly, the process $\{M_{T_m}\}_{m=0}^{M-1}$ yields a discrete martingale as

$$\mathbb{E}^{\mathbb{Q}}\left[M_{T_m}\big|\mathcal{F}_{T_{m-1}}\right] = \mathbb{E}^{\mathbb{Q}}\left[M_{T_{m-1}} + \frac{G_m(z_m(T_m))}{B(T_m)} - \mathbb{E}^{\mathbb{Q}}\left[\frac{G_m(z_m(T_m))}{B(T_m)}\bigg|\mathcal{F}_{T_{m-1}}\right]\bigg|\mathcal{F}_{T_{m-1}}\right]$$

$$= \mathbb{E}^{\mathbb{Q}}\left[M_{T_{m-1}}\big|\mathcal{F}_{T_{m-1}}\right] + \mathbb{E}^{\mathbb{Q}}\left[\frac{G_m(z_m(T_m))}{B(T_m)} - \frac{G_m(z_m(T_m))}{B(T_m)}\bigg|\mathcal{F}_{T_{m-1}}\right]$$

$$= M_{T_{m-1}}$$

Furthermore, the process $M_t$ as defined above will coincide with $Y_t$ if the approximation errors in $G_m(\cdot)$ equal zero, hence yielding an equality in Equation (11). Note that the recursive relation in Equation (12) can be rewritten as

$$M_{T_m} = \frac{G_0(z_0(T_0))}{B(T_0)} + \sum_{j=1}^{m}\left(\frac{G_j(z_j(T_j))}{B(T_j)} - \mathbb{E}^{\mathbb{Q}}\left[\frac{G_j(z_j(T_j))}{B(T_j)}\bigg|\mathcal{F}_{T_{j-1}}\right]\right) \tag{13}$$

We can now estimate the upper bound by again simulating a set of scenarios of the risk factor $\left\{\left(x_{t_1}^n, x_{t_2}^n, \ldots, x_{T_M}^n\right)\big|n = 1, \ldots, N\right\}$ and approximate $U(0)$ under the risk-neutral measure as

$$\tilde{U}(0) = M_0 + \frac{1}{N}\sum_{n=1}^{N}\max_{T_m \in \mathcal{T}_f}\left\{\frac{h_{T_m}\left(x_{T_m}^n\right)}{B^n(T_m)} - M_{T_m}^n\right\}$$

The upper bound can be approximated under the $T_M-$forward measure. In that case, the risk factor should be simulated under $\mathbb{Q}^{T_M}$ and the numéraire $B(t)$ should be replaced by $P(t, T_M)$. By carrying this out, we avoid the need to approximate the numéraire on a coarse simulation grid.

Note that by the deliberate choice of $G_m(\cdot)$, all the conditional expectations appearing in Equation (13) can be computed in closed form (see Appendix A). Hence, there is no need to resort to nested simulations, in contrast to, for example, Andersen and Broadie (2004) and Becker et al. (2020). Especially if simulations are performed under the $T_M-$forward measure, both lower and upper bound estimations can be obtained at minimal additional computational cost.

## 5. Error Analysis

In this section, we analyze the errors of the semi-static hedge, the direct estimator, the lower bound estimator, and the upper bound estimator, which are induced by the imprecision of the regression functions $G_0, \ldots, G_{M-1}$. We show that for a sufficiently large hedging portfolio, the replication error will be arbitrarily small. Furthermore, we will provide error margins for the price estimators in terms of the regression imprecision. We thereby show that the direct estimator, lower bound, and upper bound will converge to the true option price as the accuracy of the regressions increases. The cornerstone to the subsequent theorems is the universal approximation theorem, as presented in, for example, Hornik et al. (1989). Given that $\tilde{V}$ is a continuous function on the compact set $\mathcal{I}_d$, it guarantees that, for each $m \in \{0, \ldots, M-1\}$, there exists a neural network $G_m$ such that

$$\sup_{x \in \mathcal{I}_d} B^{-1}(T_m) \big| \tilde{V}(T_m; x) - G_m(z_m(T_m)|x) \big| < \varepsilon$$

for arbitrary $\varepsilon > 0$. In other words, the regression error can be kept arbitrarily small on any compact domain of the risk factor.

### 5.1. Accuracy of the Semi-Static Hedge

Let $\mathcal{T}_f = \{T_0, \ldots, T_{M-1}\}$ denote the set of monitor dates. For the following theorem, we assume that $\mathbf{x}_t \in \mathcal{I}_d$ for some compact set $\mathcal{I}_d \subset \mathbb{R}^d$. As $\mathcal{I}_d$ can be arbitrarily large, this assumption is loose enough to account for a vast majority of the risk factor scenarios in a standard Monte Carlo sample. On top of that, $\mathcal{I}_d$ can be chosen as sufficiently large such that $\mathbb{E}^{\mathbb{Q}}\Big[\big|\tilde{V}(T_m) - G_m(z_m)\big| \mathbb{1}_{\{\mathbf{x}_{T_m} \notin \mathcal{I}_d\}} \Big| \mathcal{F}_0\Big]$ approaches zero. For the proof, we refer to Appendix D.

**Theorem 1.** *Let $\varepsilon > 0$ and $|\mathcal{T}_f| = M$. Denote by $\tilde{V}(t)$ the value of the replication portfolio for a Bermudan swaption, conditional on the fact that it is not exercised prior to time $t$. Assume that there exist $M$ networks $G_m(\cdot)$ such that*

$$\sup_{x \in \mathcal{I}_d} B^{-1}(T_m) \big| \tilde{V}(T_m; x) - G_m(z_m(T_m)|x) \big| < \varepsilon, \qquad \forall_{m \in \{0, \ldots, M-1\}}$$

*Then, for any $t \in [0, T_{M-1}]$, we have that*

$$\sup_{x \in \mathcal{I}_d} B^{-1}(t) \big| V(t; x) - \tilde{V}(t; x) \big| < M\varepsilon$$

### 5.2. Error of the Direct Estimator

Theorem 1 bounds the hedging error of the semi-static hedge in terms of the maximum regression errors. This implicitly provides an error margin to the direct estimator under the aforementioned assumptions. Although the universal approximation theorem guarantees that the supremum errors can be kept at any desired level, in practice, they are substantially higher than, for example, the MSEs or MAEs of the regression function. This is due to inevitable fitting imprecision outside or near the boundaries of the finite training sets. In the following theorem, we propose that the error of the direct estimator can be bounded in terms of the discounted MAEs of the neural networks. These quantities are generally much tighter than the supremum errors and are typically easier to estimate.

The proof of the theorem follows a similar line of thought as the proof of Theorem 1. As the direct estimator at time-zero depends on the expectation of the continuation value at $T_0$, we can show by an iterative argument that the overall error is bounded by the sum of the mean absolute fitting errors at each monitor date. The error bound in the direct estimator therefore scales linearly with the number of exercise opportunities. For a complete proof, we refer to Appendix E.

**Theorem 2.** *Let $\varepsilon > 0$ and assume that $|\mathcal{T}_f| = M$. Denote by $\tilde{V}$ the time-zero direct estimator for the price of a Bermudan swaption $V$. Assume that, for each $T_m \in \{T_0, \ldots, T_{M-1}\}$, there is a neural network approximation $G_m(\cdot)$ such that*

$$\mathbb{E}^{\mathbb{Q}}\left[B^{-1}(T_m)\big|\tilde{V}(T_m) - G_m(z_m)\big|\,\Big|\,\mathcal{F}_0\right] < \varepsilon$$

*where $\tilde{V}(T_m) := \max\left\{B(T_m)\mathbb{E}^{\mathbb{Q}}\left[\frac{G_{m+1}(z_{m+1})}{B(T_{m+1})}\,\Big|\,\mathcal{F}_{T_m}\right], h_m(\mathbf{x}_{T_m})\right\}$ denotes the estimator at date $T_m$. Then, the error in $\tilde{V}$ is bounded as given below:*

$$\big|V(0) - \tilde{V}(0)\big| < M\varepsilon$$

*5.3. Tightness of the Lower Bound Estimate*

A lower bound $L(t)$ to the true price can be computed by considering the non-optimal exercise strategy, implied by the direct estimator (see Section 4.1). This relies on the stopping time

$$\tilde{\tau}(\omega) = \min\left\{T_m \in \mathcal{T}_f\big|\tilde{C}_m(T_m) \leq h_m(\mathbf{x}_{T_m})\right\} \tag{14}$$

In the following theorem, we propose that the tightness of $L(0)$ can be bounded by the discounted MAEs of neural network approximations.

The proof of the theorem relies on the fact that, conditioned on any realization of $\tilde{\tau}$ and $\tau$, the expected difference between $L(0)$ and $V(0)$ is bounded by the sum of the mean absolute fitting errors at the monitor dates between $\tilde{\tau}$ and $\tau$. In the proof, we therefore distinguish between the events $\tilde{\tau} < \tau$ and $\tilde{\tau} > \tau$. Then, by an inductive argument, we can show that the bound on the spread between $L(0)$ and the true price scales linearly with the number of exercise opportunities. For a complete proof, we refer to Appendix F.

**Theorem 3.** *Let $\varepsilon > 0$ and assume that $|\mathcal{T}_f| = M$. Denote by $L(0)$ the lower bound on the true Bermudan swaption price as defined in Equation (10). Assume that, for each $T_m \in \{T_0, \ldots, T_{M-1}\}$, there is a neural network approximation $G_m(\cdot)$, such that*

$$\mathbb{E}^{\mathbb{Q}}\left[B^{-1}(T_m)\big|\tilde{V}(T_m) - G_m(z_m)\big|\,\Big|\,\mathcal{F}_0\right] < \varepsilon$$

*where $\tilde{V}(T_m) := \max\left\{B(T_m)\mathbb{E}^{\mathbb{Q}}\left[\frac{G_{m+1}(z_{m+1})}{B(T_{m+1})}\,\Big|\,\mathcal{F}_{T_m}\right], h_m(\mathbf{x}_{T_m})\right\}$ denotes the estimator at date $T_m$. Then, the spread between $V(0)$ and $L(0)$ is bounded as given below:*

$$\big|V(0) - L(0)\big| < 2(M-1)\varepsilon$$

*5.4. Tightness of the Upper Bound Estimate*

An upper bound $U(t)$ to the true price can be computed by considering a dual formulation of the dynamic pricing equation Haugh and Kogan (2004); see Section 4.2. From a practical point of view, the difference between the upper bound and the true price can be interpreted as the maximum loss that an investor would incur due to hedging imprecision resulting from the algorithm Lokeshwar et al. (2022). The overall hedging error at some monitor date $T_m$ is the result of all incremental hedging errors occurring from rebalancing the portfolio at preceding monitor dates. As the incremental hedging errors can be bounded by the sum of the expected absolute fitting errors, we propose that the tightness of $U(t)$ can be bounded by the discounted MAEs of the neural networks and scales at most quadratically with the number of exercise opportunities.

The proof follows a similar line of thought as that presented in Andersen and Broadie (2004). There, it is noted that the difference between the dual formulation of the option and its true price is difficult to be bound. Here, we make a similar remark and propose a

theoretical maximum spread between $U(0)$ and $V(0)$ that is relatively loose. Our numerical experiments, however, indicate that the upper bound estimate is much tighter in practice. For a complete proof, we refer to Appendix G.

**Theorem 4.** *Let $\varepsilon > 0$ and assume that $|\mathcal{T}_f| = M$. Denote by $U(0)$ the upper bound on the true Bermudan swaption price as defined in Equation (11). Assume that, for each $T_m \in \{T_0, \dots, T_{M-1}\}$, there is a neural network approximation $G_m(\cdot)$, such that*

$$\mathbb{E}^{\mathbb{Q}}\left[B^{-1}(T_m)\left|\tilde{V}(T_m) - G_m(z_m)\right|\,\middle|\,\mathcal{F}_0\right] < \varepsilon$$

*where $\tilde{V}(T_m) := \max\left\{B(T_m)\mathbb{E}^{\mathbb{Q}}\left[\frac{G_{m+1}(z_{m+1})}{B(T_{m+1})}\middle|\mathcal{F}_{T_m}\right], h_m(\mathbf{x}_{T_m})\right\}$ denotes the estimator at date $T_m$. Then, the spread between $V(0)$ and $U(0)$ is bounded as given below:*

$$|U(0) - V(0)| < M(M-1)\varepsilon$$

## 6. Numerical Experiments

In this section, we treat several numerical examples to illustrate the convergence, pricing, and hedging performance of our proposed method. We will start by considering the price estimate of a vanilla swaption contract in a one-factor model. This is a toy example by which we can demonstrate the accuracy of the direct estimator in comparison to exact benchmarks. We continue with price estimates of Bermudan swaption contracts in a one-factor and a two-factor framework. The performance of the direct estimator will be compared to the established least-square regression method (LSM) introduced in Longstaff and Schwartz (2001), fine-tuned to an interest rate setting as described in Oosterlee et al. (2016). Additionally, we will approximate the lower and upper bound estimates as described in Section 4 and show that they are well inside the error margins introduced in Section 5. Finally, we will illustrate the performance of the static hedge for a swaption in a one-factor model and a Bermudan swaption in a two-factor model. For the one-factor case, we can benchmark the performance by the analytic delta hedge for a swaption, provided in Henrard (2003).

A $T_0 \times T_M$ contract (either European swaption or Bermudan swaption) refers to an option written on a swap with a notional amount of 100 and a lifetime between $T_0$ and $T_M$. This means that $T_0$ and $T_{M-1}$ are the first and last monitor dates, respectively, in case of a Bermudan. The underlying swaps are set to exchange annual payments, yielding year fractions of 1 and annual exercise opportunities. All examples that are illustrated here have been implemented in Python using the Quant-Lib library Ametrano and Ballabio (2003) for standard pricing routines and Keras with Tensorflow backend Chollet et al. (2015) for constructing, fitting, and evaluating the neural networks.

### 6.1. 1-Factor Swaption

We start by considering a swaption contract under a one-dimensional risk factor setting. The direct estimator of the true $V(0)$ swaption price is computed similar to a Bermudan swaption, but with only a single exercise possibility at $T_0$. Therefore, only a single neural network per option needs to be trained to compute the option price. We have used 64 hidden nodes and 20,000 training points, generated through Monte Carlo sampling. We assume the risk factor to be captured by the Hull–White model with constant mean reversion parameter $a$ and constant volatility $\sigma$. The dynamics of the shifted mean-zero process Brigo and Mercurio (2006) are hence given by

$$dx(t) = -ax(t)dt + \sigma dW(t), \qquad x(0) = 0 \tag{15}$$

For simplicity, we consider a flat time-zero instantaneous forward rate $f(0, t)$. The risk-neutral scenarios are generated using a discrete Euler scheme of the process above. Parameter values that were used in the numerical experiments are summarized in Table 1.

**Table 1.** Parameters 1F Hull–White model.

| Parameter | $a$ | $\sigma$ | $f(0, t)$ |
|:---:|:---:|:---:|:---:|
| **Value** | 0.01 | 0.01 | 0.03 |

Figure 3a,b show the time-zero option values in basis points (0.01%) of the notional for a $5Y \times 10Y$ and a $10Y \times 5Y$ payer swaption as a function of the moneyness. The moneyness is defined as $\frac{S}{K}$, where $K$ denotes the fixed strike and $S$ the time-zero swap rate associated with the underlying swap. The exact benchmarks are computed by an application of Jamshidian's decomposition Jamshidian (1989). The relative estimate errors are shown in Figure 3c,d. We observe a close agreement between the estimates and the reference prices. The errors are in the order of several basis points of the true option price. In the current setting, the results presented serve mostly as a validation of the estimator. We however point out that this algorithm for swaptions is applicable in general frameworks, such as multi-factor, dual-curve, or non-overlapping payment schemes, for which exact routines are no longer available.

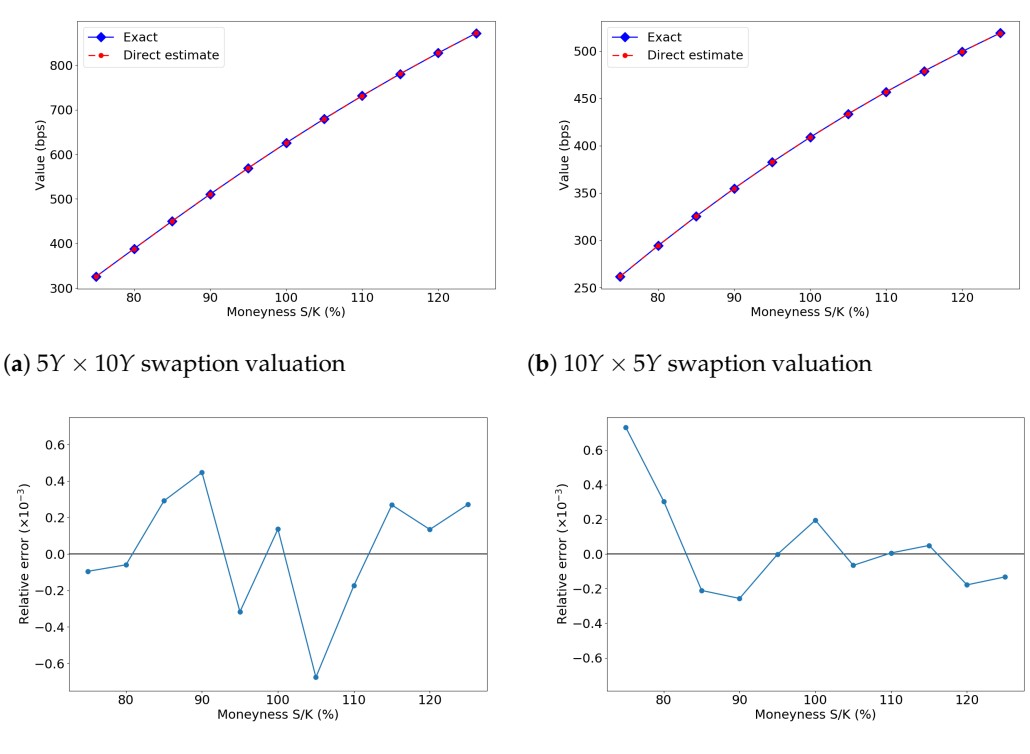

(**a**) $5Y \times 10Y$ swaption valuation

(**b**) $10Y \times 5Y$ swaption valuation

(**c**) Error $5Y \times 10Y$ swaption estimates

(**d**) Error $10Y \times 5Y$ swaption estimates

**Figure 3.** Accuracy of the direct estimator for vanilla swaptions. $S_{5Y \times 10Y} \approx S_{10Y \times 5Y} \approx 0.0305$.

*6.2. 1-Factor Bermudan Swaption*

As a second example, we consider a Bermudan swaption contract. The same dynamics for the underlying risk factor are assumed as discussed in the previous paragraph, using the parameter settings of Table 1. Monte Carlo scenarios are generated based on a discretized Euler scheme associated to the SDE in Equation (15), taking weekly time-steps.

We first demonstrate the convergence property of the direct estimator, which is implied by the replication portfolio. We consider a $1Y \times 5Y$ Bermudan swaption with strike $K = 0.03$. This strike is selected as it close to ATM, a moneyness level that is most likely to be liquid

in the market. For this analysis, the neural networks were trained to a set of 2000 Monte-Carlo-generated training points. Figure 4a shows the direct estimator as a function of the number of hidden nodes in each neural network, alongside an LSM-based benchmark. In Figure 4b, the error with respect to the LSM estimate is shown on a logscale. We observe that the direct estimator converges to the LSM confidence interval or slightly above, which is in accordance with the fact that LSM is biased low by definition. The analysis indicates that a portfolio of 16 discount bond options is sufficient to achieve a replication of a similar accuracy to the LSM benchmark.

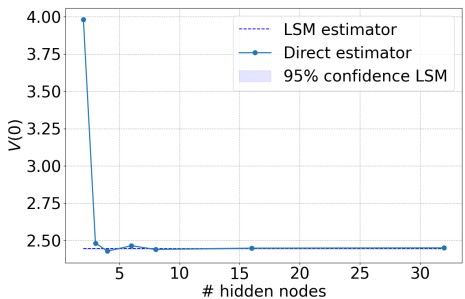

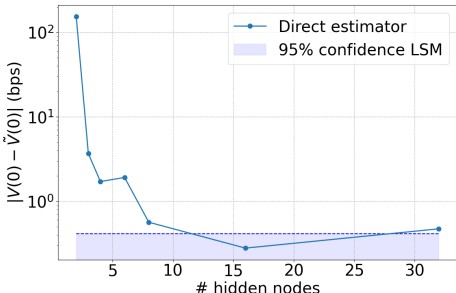

(**a**) Convergence price

(**b**) Convergence pricing error

**Figure 4.** Convergence of the direct estimator for the $1Y \times 5Y$ Bermudan swaption price as a function of hidden node count, with respect to the LSM benchmark under a 1-factor model.

Table 2 depicts numerical pricing results for a $1Y \times 5Y$, $3Y \times 7Y$ and $1Y \times 10Y$ receiver Bermudan swaption. For each contract, we consider different levels of moneyness, setting the fixed rate $K$ of the underlying swap to, respectively, 80%, 100%, and 120% of the time-zero swap rate. The estimations of the direct, the upper bound, and the lower bound statistics are again reported alongside LSM-based benchmarks. Here, the neural networks have 64 hidden nodes and are fitted using a training set of 20,000 points. The lower and upper bound estimates, as well as the LSM estimates, are based on simulation runs of 200,000 paths each. The given lower and upper bounds are Monte Carlo estimates of the statistics defined in Equations (10) and (11) and are therefore subject to standard errors, which are reported in parentheses. The reference LSM results have been generated using $\{1, x, x^2\}$ as regression basis functions for approximating the continuation values. The standard errors and confidence intervals are obtained from ten independent Monte Carlo runs. The choice for hyperparameter settings is motivated by the analysis of Appendix C.

**Table 2.** Results of 1-factor model. $S_{1Y \times 5Y} \approx S_{3Y \times 7Y} \approx S_{1Y \times 10Y} \approx 0.0305$. Standard errors are in parentheses, based on 10 independent MC runs of $2 \times 10^5$ paths each.

| Type | K/S | Dir.est. | Lower bnd | Upper bnd | UB-LB | LSM est. | LSM 95% CI |
|------|-----|----------|-----------|-----------|-------|----------|------------|
| $1Y \times 5Y$ | 80% | 1.527 | 1.521 (0.001) | 1.528 (0.000) | 0.007 | 1.521 (0.001) | [1.518, 1.523] |
| | 100% | 2.543 | 2.534 (0.002) | 2.542 (0.000) | 0.008 | 2.534 (0.002) | [2.531, 2.538] |
| | 120% | 4.015 | 4.016 (0.002) | 4.018 (0.000) | 0.002 | 4.016 (0.002) | [4.012, 4.021] |
| $3Y \times 7Y$ | 80% | 3.296 | 3.293 (0.002) | 3.295 (0.000) | 0.002 | 3.293 (0.002) | [3.290, 3.296] |
| | 100% | 4.767 | 4.755 (0.004) | 4.761 (0.000) | 0.006 | 4.755 (0.004) | [4.747, 4.762] |
| | 120% | 6.625 | 6.629 (0.004) | 6.631 (0.000) | 0.002 | 6.629 (0.004) | [6.621, 6.638] |
| $1Y \times 10Y$ | 80% | 3.950 | 3.945 (0.005) | 3.960 (0.000) | 0.015 | 3.945 (0.005) | [3.935, 3.955] |
| | 100% | 5.818 | 5.811 (0.003) | 5.818 (0.000) | 0.007 | 5.811 (0.003) | [5.805, 5.816] |
| | 120% | 8.346 | 8.354 (0.005) | 8.360 (0.000) | 0.006 | 8.353 (0.005) | [8.344, 8.362] |

The spreads between the lower and upper bound estimates provide a good indication of the accuracy of the method. For the current setting, we obtain spreads in the order of several basis points up a few dozen of basis points. The lower bound estimate is typically very close to the LSM estimate, which itself is also biased low. Their standard errors are of the same order of magnitude. The upper bound estimates prove to be very stable and show a variance that is roughly two orders of magnitude smaller compared to that of the lower bound. The direct estimate is occasionally slightly less accurate. This can be explained by the fact that it depends on the accuracy of the regression over the full domain of the risk factor, whereas, for the lower bound, only a high accuracy near the exercise boundaries is required. In Figure 5, the mean absolute error of each neural network after fitting is presented as a function of the network's index. The errors are displayed in basis points of the notional. We observe that the errors are the smallest at maturity and tend to increase with each iteration backward in time. That the errors at the final monitor date are virtually zero can be explained by the fact that the pay-off at $T_{M-1}$ is given by

$$\max\{h_{M-1}(\mathbf{x}_{T_{M-1}}), 0\} = N \cdot \max\{A_{M-1,M}(T_{M-1}) \cdot (K - S_{M-1,M}(T_{M-1})), 0\}$$
$$= N \cdot \max\{(\Delta T_M K + 1)P(T_{M-1}, T_M) - 1, 0\}$$
$$\simeq w_2 \varphi(w_1 z - b)$$

which can be exactly captured by a network with only a single hidden node. With each step backwards, the target function is harder to fit, yielding larger errors. We observe MAEs up to one basis point of the notional amount. The empirical lower–upper bound spreads remain well within the theoretical error margins provided in Sections 4.1 and 4.2. The spreads are mostly much lower than the sum of the MAEs, indicating that the bound estimates are in practice significantly tighter than their theoretical maximum spread.

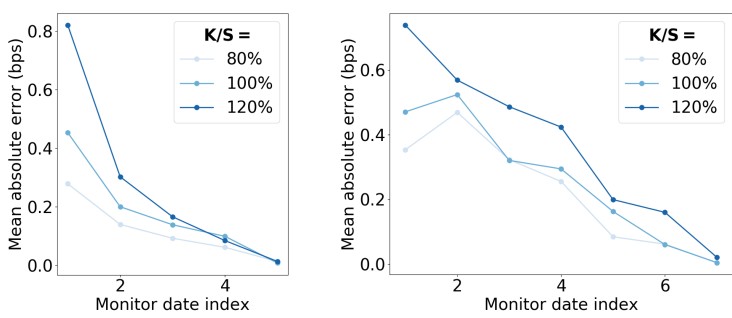

(**a**) 1Y×5Y Bermudan      (**b**) 3Y×7Y Bermudan

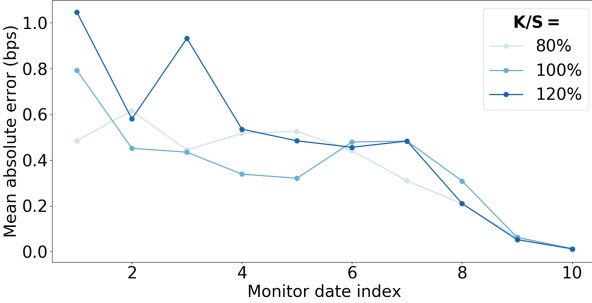

(**c**) 1Y×10Y Bermudan

**Figure 5.** Mean absolute errors of neural network fit per monitor date under a 1-factor model.

*6.3. 2-Factor Bermudan Swaption*

As a final pricing example, we consider a Bermudan swaption contract under a two-factor model. The dynamics of the underlying risk factors are assumed to follow a

G2++ model Brigo and Mercurio (2006). Monte Carlo scenarios are generated based on a discretized Euler scheme, taking weekly time-steps, based on the SDE below:

$$dx_1(t) = -a_1 x_1(t)dt + \sigma_1 dW_1(t), \qquad x_1(0) = 0$$
$$dx_2(t) = -a_2 x_2(t)dt + \sigma_2 dW_2(t), \qquad x_2(0) = 0$$

where $W_1$ and $W_2$ are correlated Brownian motions with $d\langle W_1, W_2\rangle_t = \rho dt$. Parameter values that were used in the numerical experiments are summarized in Table 3.

We again start by demonstrating the convergence property of the direct estimator for both the locally connected and the fully connected neural network designs as specified in Section 3.2.3. The same $1Y \times 5Y$ Bermudan swaption with strike $K = 0.03$ is used and the networks are each fitted to a set of 6400 training points. Figure 6a shows the direct estimator as a function of the number of hidden nodes in each neural network, alongside an LSM-based benchmark. In Figure 6b, the error with respect to the LSM estimate is shown on a logscale. We observe a similar convergence behavior, where the direct estimators approach the LSM benchmark within the 95% confidence range. Here, it is noted that a portfolio of eight discount bond options is already sufficient to achieve a replication of a similar accuracy to the LSM estimator.

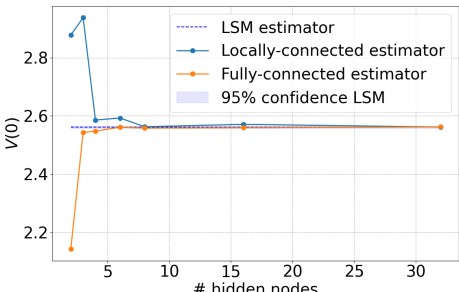
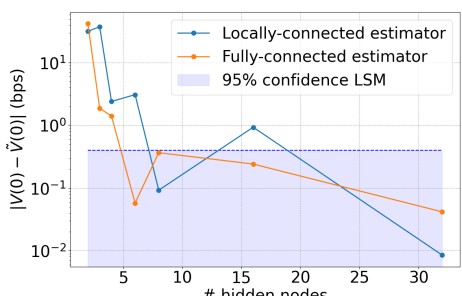

(**a**) Convergence price

(**b**) Convergence pricing error

**Figure 6.** Convergence of the direct estimator for the $1Y \times 5Y$ Bermudan swaption price as a function of hidden node count, with respect to the LSM benchmark under a 2-factor model.

**Table 3.** Parameters 2F G2++ model.

| Parameter | $a_1$ | $a_2$ | $\sigma_1$ | $\sigma_2$ | $\rho$ | $f(0,t)$ |
|---|---|---|---|---|---|---|
| **Value** | 0.07 | 0.08 | 0.015 | 0.008 | $-0.6$ | 0.03 |

In Table 4, numerical results for a $1Y \times 5Y$, $3Y \times 7Y$, and $1Y \times 10Y$ receiver Bermudan swaption are depicted for different levels of moneyness. We again report the direct, the upper bound, and the lower bound estimates for both neural network designs. In this case, all networks have 64 hidden nodes and are fitted to training sets of 20,000 points. As before, the lower bound, the upper bound, and the LSM estimates are the result of 10 independent Monte Carlo simulations of 200,000 scenarios.

For the LSM algorithm, we used $\{1, x_1, x_2, x_1^2, x_1 x_2, x_2^2\}$ as basis functions. Note that the number of monomials grows quadratically with the dimension of the state space and, with that, the number of free parameters. For our method, this number grows at a linear rate. Choices for the hyperparameters are again based on the analysis of Appendix C. The results under the two-factor case share several features with the one-factor results. We observe spreads between the lower and upper bounds ranging from several basis points up to a few dozen basis points of the option price. The lower bound estimates turn out to be very close to the LSM estimates and the same holds for their standard errors. The upper bounds are again very stable with low standard errors and the direct estimator appears as slightly less accurate. If we compare the locally connected to the fully connected case, we observe that the results are overall in close agreement, especially the lower and upper

bound estimates. This is remarkable given that the fully connected case gives rise to more trainable parameters, by which we would expect a higher approximation accuracy. In the two-factor setting, the ratio of free parameters for the two designs is 3:4.

**Table 4.** Results of 2-factor model for the locally connected and fully connected neural network cases. $S_{1Y \times 5Y} \approx S_{3Y \times 7Y} \approx S_{1Y \times 10Y} \approx 0.0305$. Standard errors are in parentheses, based on 10 independent MC runs of $2 \times 10^5$ paths each.

| | | | LOCALLY CONNECTED NEURAL NETWORKS | | | | |
|---|---|---|---|---|---|---|---|
| Type | K/S | Dir.est. | Lower bnd | Upper bnd | UB-LB | LSM est. | LSM 95% CI |
| $1Y \times 5Y$ | 80% | 1.617 | 1.617(0.002) | 1.619(0.000) | 0.002 | 1.617(0.002) | [1.614, 1.621] |
| | 100% | 2.652 | 2.650(0.002) | 2.654(0.000) | 0.004 | 2.650(0.002) | [2.646, 2.654] |
| | 120% | 4.128 | 4.127(0.003) | 4.131(0.000) | 0.004 | 4.127(0.003) | [4.121, 4.132] |
| $3Y \times 7Y$ | 80% | 3.073 | 3.076(0.004) | 3.078(0.000) | 0.002 | 3.077(0.004) | [3.069, 3.085] |
| | 100% | 4.554 | 4.553(0.004) | 4.553(0.000) | 0.000 | 4.552(0.004) | [4.545, 4.559] |
| | 120% | 6.444 | 6.448(0.004) | 6.451(0.000) | 0.003 | 6.446(0.005) | [6.435, 6.456] |
| $1Y \times 10Y$ | 80% | 3.616 | 3.624(0.002) | 3.626(0.000) | 0.002 | 3.622(0.002) | [3.618, 3.627] |
| | 100% | 5.508 | 5.509(0.002) | 5.514(0.000) | 0.005 | 5.508(0.002) | [5.503, 5.512] |
| | 120% | 8.128 | 8.123(0.005) | 8.130(0.000) | 0.007 | 8.121(0.005) | [8.110, 8.132] |
| | | | FULLY CONNECTED NEURAL NETWORKS | | | | |
| Type | K/S | Dir.est. | Lower bnd | Upper bnd | UB-LB | LSM est. | LSM 95% CI |
| $1Y \times 5Y$ | 80% | 1.617 | 1.617(0.002) | 1.619(0.000) | 0.002 | 1.617(0.002) | [1.614, 1.621] |
| | 100% | 2.651 | 2.650(0.002) | 2.654(0.000) | 0.004 | 2.650(0.002) | [2.646, 2.654] |
| | 120% | 4.129 | 4.127(0.003) | 4.131(0.000) | 0.004 | 4.127(0.003) | [4.121, 4.132] |
| $3Y \times 7Y$ | 80% | 3.076 | 3.077(0.004) | 3.078(0.000) | 0.001 | 3.077(0.004) | [3.069, 3.085] |
| | 100% | 4.553 | 4.553(0.004) | 4.554(0.000) | 0.001 | 4.552(0.004) | [4.545, 4.559] |
| | 120% | 6.451 | 6.447(0.005) | 6.451(0.000) | 0.004 | 6.446(0.005) | [6.435, 6.456] |
| $1Y \times 10Y$ | 80% | 3.616 | 3.624(0.002) | 3.626(0.000) | 0.002 | 3.622(0.002) | [3.618, 3.627] |
| | 100% | 5.506 | 5.509(0.002) | 5.514(0.000) | 0.005 | 5.508(0.002) | [5.503, 5.512] |
| | 120% | 8.124 | 8.123(0.005) | 8.130(0.000) | 0.007 | 8.121(0.005) | [8.110, 8.132] |

In Figure 7, the mean absolute errors of the neural networks after fitting are shown. The MAEs for the locally connected networks are in blue; the fully connected are in red. All are represented in basis points of the notional amount. We observe that the errors are mostly in the same order of magnitude as the one-dimensional case. The figures indicate that the locally connected networks slightly outperform the fully connected networks in terms of accuracy, although this does not appear to materialize in tighter estimates of the lower and upper bounds. For the locally connected case, we again observe that the errors are virtually zero at the last monitor date, for the same reasons as in the one-factor setting. In the fully connected representation, an exact replication might not exist, resulting in larger errors. We conjecture that this effect partially carries over to the networks at preceding monitor dates. The empirical lower–upper bound spreads remain well within the theoretical error margins, as the spreads are in all cases lower than the sum of the MAEs. Hence, also for the two-factor setting, we find that the bound estimates are tighter in practice than their theoretical maximum spreads.

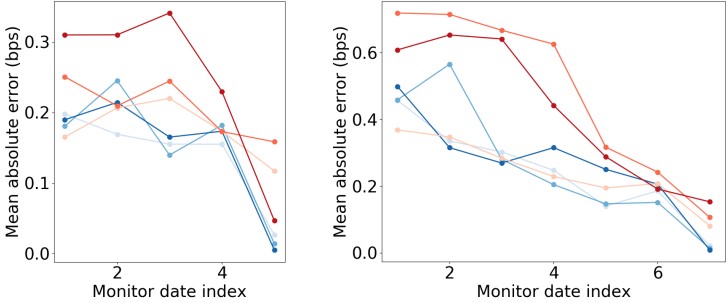

(**a**) 1Y×5Y Bermudan  (**b**) 3Y×7Y Bermudan

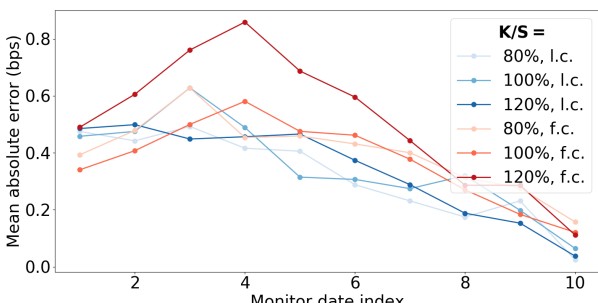

(**c**) 1Y×10Y Bermudan

**Figure 7.** Accuracy of neural network fit per monitor date under a 2-factor model. Blue lines represent the locally connected (l.c.) case and the red lines represent the fully connected (f.c.) case. The legend in Figure (**c**) applies to all three graphs.

### 6.4. Performance Semi-Static Hedge

Finally, we consider the hedging problem of a vanilla swaption under the one-factor model and a Bermudan swaption under the two-factor model.

#### 6.4.1. 1-Factor Swaption

Here, we compare the performance of a static hedge versus a dynamic hedge in the one-factor model. As an example, we take a $1Y \times 5Y$ European receiver swaption at different levels of moneyness. The model set-up is similar to that in Section 6.2, using the same set of parameters reported in Table 1. In the static hedge case, the option contract writer aims to hedge the risk using a static portfolio of zero-coupon bond options and discount bonds. The replicating portfolio is composed using a neural network with 64 hidden nodes, optimized using 20,000 training-points generated through Monte Carlo sampling. The portfolio is composed at time-zero and kept until the expiry of the option at $t = 1$ year. In the dynamic hedge case, the delta-hedging strategy is applied. The replicating portfolio is composed of units of the underlying forward-starting swap and investment in the money market. The dynamic hedge involves the periodic rebalancing of the portfolio. The delta for a receiver swaption under the Hull–White model (see Henrard 2003) is given by

$$\Delta(t) = \frac{\sum_{j=1}^{M} c_j P(t, T_j) \nu(t, T_j) \Phi(\kappa + \alpha_j) - P(t, T_0) \nu(t, T_0) \Phi(\kappa)}{\sum_{j=1}^{M} c_j P(t, T_j) \nu(t, T_j) - P(t, T_0) \nu(t, T_0)} \tag{16}$$

where $\kappa$ is the solution of

$$\sum_{j=1}^{M} c_j \frac{P(t, T_j)}{P(t, T_0)} \exp\left(-\frac{1}{2}\alpha_j^2 - \alpha_j \kappa\right) = 1$$

and

$$\alpha_j^2 := \int_0^{T_0} \left(\nu(u, T_j) - \nu(u, T_0)\right)^2 du$$

where $\Phi$ denotes the CDF of a standard normal distribution, $c_j = \Delta T_j K$ for $j = 1, \ldots, M-1$, and $c_M = 1 + \Delta T_M K$. The function $\nu(t, T)$ denotes the instantaneous volatility of a discount bond maturing at $T$, which, under Hull–White, is given by $\nu(t, T) := \frac{\sigma}{a}\left(1 - e^{-a(T-t)}\right)$. We validated the analytic expression above with numerical approximations of the Delta obtained by bumping the yield curve. Within the simulation, the dynamic hedge portfolio is rebalanced on a daily basis between time-zero and expiry of the option. In this experiment, that means it is updated on 255 instances at equidistant monitor dates.

The performance of both hedging strategies is reported in Table 5. The results are based on 10,000 risk-neutral Monte Carlo paths. The hedging error refers to the difference between the option's pay-off at expiry and the replicating portfolio's final value. The quantities are reported in basis points of the notional amount. The empirical distribution of the hedging error is shown in Figure 8. We observe that, overall, the static hedge outperforms the dynamic hedge in terms of accuracy, even though it involves only a quarter (64 versus 255) of the trades. Although it is not visible in Figure 8b, the static strategy does give rise to occasional outliers in terms of accuracy. These are associated with scenarios that reach or exceed the boundary of the training set. These errors are typically of a similar order of magnitude as the errors observed in the dynamic hedge. The impact of outliers can be reduced by increasing the training set and thereby broadening the regression domain.

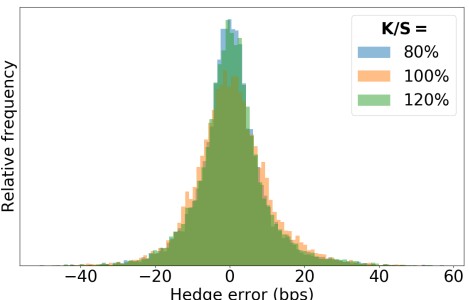
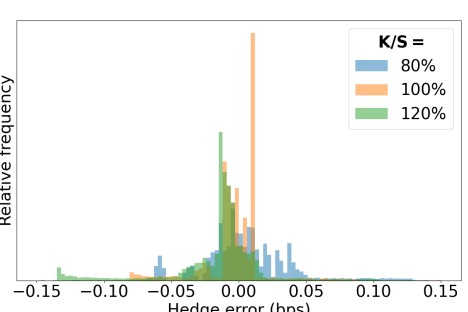

(**a**) Hedge error dynamic strategy

(**b**) Hedge error static strategy

**Figure 8.** Hedge error distribution for a $1Y \times 5Y$ receiver swaption, based on $10^4$ MC paths. $S_{1Y \times 5Y} \approx 0.0305$.

**Table 5.** Hedging errors for static and dynamic hedging strategy for a $1Y \times 5Y$ receiver swaption, based on $10^4$ MC paths. $S_{1Y \times 5Y} \approx 0.0305$.

| Hedge Error (bps) | K/S | Static Hedge | Dyn. Hedge |
|---|---|---|---|
| Mean | 80% | $-1.9 \times 10^{-2}$ | 0.38 |
| | 100% | $-2.2 \times 10^{-3}$ | 0.61 |
| | 120% | $-1.5 \times 10^{-2}$ | 0.46 |
| St. dev. | 80% | 2.5 | 9.1 |
| | 100% | $3.1 \times 10^{-2}$ | 10.1 |
| | 120% | $4.5 \times 10^{-2}$ | 9.4 |
| 95%-percentile | 80% | $6.6 \times 10^{-2}$ | 15.7 |
| | 100% | $1.2 \times 10^{-2}$ | 17.9 |
| | 120% | $2.0 \times 10^{-2}$ | 16.2 |

6.4.2. 2-Factor Bermudan Swaption

Here, we demonstrate the performance of the semi-static hedge for a $1Y \times 5Y$ receiver Bermudan swaption under a two-factor model. We compare the accuracy of the hedging strategy utilizing a locally connected network versus a fully connected neural network.

In the former, the replication portfolio consists of zero-coupon bonds and zero-coupon bond options. In the latter, the Bermudan is replicated with options written on hypothetical assets with a pay-off equal to the log of a zero-coupon bond (see Section 3.2.3). The model set-up is similar to that in Section 6.3, using the same set of parameters reported in Table 3. Both networks are composed with 64 hidden nodes and optimized using 20,000 training points generated through Monte Carlo sampling. The portfolio is set up at time-zero and updated at each monitor date of the Bermudan until it is either exercised or expired. We assume that the holder of the Bermudan swaption follows the exercise strategy implied by the algorithm, i.e., the option is exercised as soon as $\tilde{C}_m(T_m) \leq h_m(\mathbf{x}_{T_m})$. When a monitor date $T_m$ is reached, the replication portfolio matures with a pay-off equal to $G_m(z_m(T_m))$. In case the Bermudan is continued, the price to set up a new replication portfolio is given by $\tilde{V}(T_m) = B(T_m)\mathbb{E}^{\mathbb{Q}}\left[\frac{G_{m+1}(z_{m+1})}{B(T_{m+1})}\Big|\mathcal{F}_{T_m}\right]$, which contributes $G_m(z_m(T_m)) - \tilde{V}(T_m)$ to the hedging error. In case the Bermudan is exercised, the holder will claim $\tilde{V}(T_m) = h_m(\mathbf{x}_{T_m})$, which also contributes $G_m(z_m(T_m)) - \tilde{V}(T_m)$ to the hedging error. The total error of the semi-static hedge (HE) is therefore computed as

$$\text{HE} := \sum_{m=0}^{M-1} \left(G_m(z_m(T_m)) - \tilde{V}(T_m)\right)\mathbb{1}_{\{\tilde{\tau} \leq T_m\}}$$

where $\tilde{V}(T_m) := \max\left\{B(T_m)\mathbb{E}^{\mathbb{Q}}\left[\frac{G_{m+1}(z_{m+1})}{B(T_{m+1})}\Big|\mathcal{F}_{T_m}\right], h_m(\mathbf{x}_{T_m})\right\}$ denotes the direct estimator at date $T_m$ and $\tilde{\tau}$ denotes the stopping time, as defined in Equation (9).

The performance of the strategies related to locally and fully connected neural networks is reported in Table 6. The results are based on 10,000 risk-neutral Monte Carlo paths and reported in basis points of the notional amount. The empirical distribution of the hedging error is shown in Figure 9. We observe that both approaches yield an accuracy in the same order of magnitude, although the locally connected case slightly outperforms the fully connected case. This is in line with expectations, as the fitting performance of the locally connected networks is generally higher. For similar reasons to the one-factor case, the hedging experiments give rise to occasional outliers in terms of accuracy. These outliers can be in the order of several dozens of basis points. Again, the impact of outliers can be reduced by broadening the regression domain.

**Table 6.** Hedging errors of the semi-static hedging strategy for a $1Y \times 5Y$ receiver Bermudan swaption, based on $10^4$ MC paths. $S_{1Y \times 5Y} \approx 0.0305$.

| Hedge Error (bps) | K/S | Loc. conn. NN | Fully conn. NN |
|---|---|---|---|
| Mean | 80% | $3.2 \times 10^{-2}$ | $2.1 \times 10^{-2}$ |
| | 100% | $7.9 \times 10^{-2}$ | $-5.5 \times 10^{-2}$ |
| | 120% | $-9.4 \times 10^{-2}$ | $4.5 \times 10^{-2}$ |
| St. dev. | 80% | 0.45 | 0.55 |
| | 100% | 0.38 | 0.48 |
| | 120% | 0.37 | 0.67 |
| 95%-percentile | 80% | 0.66 | 0.69 |
| | 100% | 0.56 | 0.85 |
| | 120% | 0.72 | 0.76 |

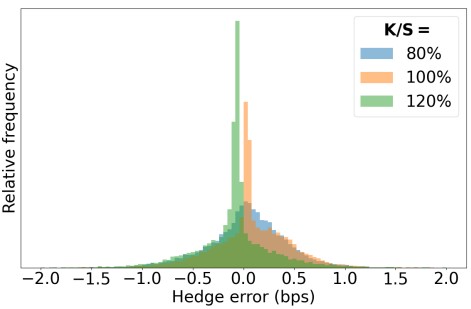
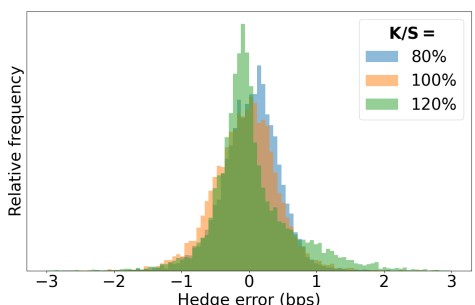

(**a**) Hedge error locally connected NN     (**b**) Hedge error fully connected NN

**Figure 9.** Hedge error distribution for a $1Y \times 5Y$ receiver Bermudan swaption, based on $10^4$ MC paths. $S_{1Y \times 5Y} \approx 0.0305$.

## 7. Conclusions

In this paper, we have proposed a semi-static replication algorithm for Bermudan swaptions under an affine term structure model. We have shown that Bermudan swaptions, an exotic interest rate derivative that is heavily traded in the OTC market, can be semi-statically replicated with an options portfolio written on a basket of discount bonds. The static portfolio composition is obtained by regressing the target option's value using a shallow, artificial neural network. The choice of the regression basis functions are motivated by their representation of an option's portfolio pay-off, implying an interpretable neural network structure. Leveraging the approximating power of ANNs, we proved that the replication can achieve any desired level of accuracy given that the portfolio is sufficiently large. We derived a direct estimator of the contract price, and an upper bound and lower bound estimate to this price can be computed at minimal additional computational cost.

The algorithm we presented is inspired by the work of Lokeshwar et al. (2022), which proposes a semi-static replication approach for callable equity options embedded in the Black–Scholes model. We contribute to the literature by extending the concept of (semi-)static replication to the field of interest rate modeling. Next, to a direct, lower bound, and upper bound estimator, we have derived analytical error margins for these statistics. This proves their convergence as the regression error diminishes and provides a direct insight toward the accuracy of the estimates. Additionally, we propose an alternative ANN design, which constrains the replication into a portfolio of vanilla bond options, even in the case of a multi-factor model. This guarantees efficiency in the portfolio valuation, which is key to many applications in credit risk management.

The performance of the method was demonstrated through several numerical experiments. We focused on Bermudan swaptions under a one- and two-factor model, which are popular amongst practitioners. The pricing accuracy of the method was determined through a benchmark to the established least-square method of Longstaff and Schwartz (2001). This reference is approached with basis point precision. A convergence analysis showed that a portfolio of 16 bond options suffices in achieving a replication with a similar accuracy to the LSM. Finally, the replication performance was studied through an in-model hedging experiment. This showed that the semi-static hedge outperforms a traditional dynamic replication in terms of hedging error.

As a look-out for further research, we consider applying the algorithm to the computation of credit risk measures and various value adjustments (xVAs). These metrics typically rely on generating forward value and sensitivity profiles of (exotic) derivative portfolios. We see the semi-static replication approach combined with the simple error analysis as an effective tool to address the computational challenges associated with these risk measures. The performance of the method in the context of quantifying CCR will therefore be studied in a forthcoming companion paper.

**Author Contributions:** Conceptualization, J.H., S.J. and D.K.; Formal analysis, J.H., S.J., D.K.; Investigation, J.H.; Writing—original draft, J.H.; Writing—review and editing, S.J. and D.K.; Visualization, J.H.; Supervision, S.J. and D.K.; Project administration, D.K. All authors have read and agreed to the published version of the manuscript.

**Funding:** This project has received funding from the NWO under the Industrial Doctorates grant. Grant Number: NWA.ID.17.029

**Data Availability Statement:** No new data were created or analyzed in this study. Data sharing is not applicable to this article.

**Conflicts of Interest:** The authors declare no conflict of interest.

**Disclosure:** The opinions expressed in this work are solely those of the authors and do not represent in any way those of their current and past employers.

## Appendix A. Evaluation of the Conditional Expectation

In this section, we will explicitly compute the conditional expectations related to the continuation values. We will distinguish two approaches associated with the two proposed network structures, i.e., the locally connected case (suggestion 1) and the fully connected case (suggestion 2).

For ease of computation, we will use a simplified, yet equivalent representation of the risk factor dynamics discussed in Section 2.1. This concerns a linear shift of the canonical representation of the latent factors as presented in Dai and Singleton (2000). We write $\mathbf{x}_t := (x_1(t), \ldots, x_n(t))^\top$, where each component $x_i$ denotes a mean-reverting zero-mean process. The risk-neutral dynamics are assumed to satisfy

$$d\begin{pmatrix} x_1(t) \\ \vdots \\ x_d(t) \end{pmatrix} = -\begin{pmatrix} a_1(t)x_1(t) \\ \vdots \\ a_d(t)x_d(t) \end{pmatrix} dt + \begin{pmatrix} \sigma_{11}(t) & \cdots & \sigma_{1d}(t) \\ \vdots & \ddots & \vdots \\ \sigma_{d1}(t) & \cdots & \sigma_{dd}(t) \end{pmatrix} d\mathbf{W}(t), \quad \begin{pmatrix} x_1(0) \\ \vdots \\ x_d(0) \end{pmatrix} = \begin{pmatrix} 0 \\ \vdots \\ 0 \end{pmatrix} \tag{A1}$$

where $\mathbf{W}$ denotes a standard $d$-dimensional Brownian motion with independent entries. By setting $\tilde{\sigma}_i(t) := \sqrt{\sum_{j=1}^d \sigma_{ij}^2(t)}$, the process above can be rewritten in terms of one-dimensional Itô processes Shreve (2004) of the form

$$dx_i(t) = -a_i(t)x_i(t)dt + \tilde{\sigma}_i(t)d\tilde{W}_i(t), \qquad i = 1, \ldots, d \tag{A2}$$

where $\tilde{W}_1, \ldots, \tilde{W}_d$ denote a set of one-dimensional, correlated Brownian motions under the measure $\mathbb{Q}$. The instantaneous correlation is denoted by $\rho_{ij}$, such that $d\langle \tilde{W}_i, \tilde{W}_j \rangle_t = \rho_{ij}(t)dt$.

### Appendix A.1. The Continuation Value with Locally Connected NN

We consider the network $G_m(\cdot)$, which is trained to approximate $\tilde{V}(T_m)$. Let $t \in [T_{m-1}, T_m)$. In order to obtain $\tilde{V}(t)$, we need to evaluate $\mathbb{E}^{\mathbb{Q}}\left[ e^{-\int_t^{T_m} r(u)du} G_m(\mathbf{x}_{T_m}) \Big| \mathcal{F}_t \right]$. As $G_m(\cdot)$ represents the linear combination of the outcome of $q$ hidden nodes, we will focus on the conditional expectation of hidden node $i \in \{1, \ldots, q\}$. Our aim is then to compute the following:

$$H_i(t) := \mathbb{E}^{\mathbb{Q}}\left[ e^{-\int_t^{T_m} r(u)du} \varphi(\mathbf{w}_i^\top \mathbf{P}(T_m) + b_i) \Big| \mathcal{F}_t \right]$$

The map $\varphi : \mathbb{R} \to \mathbb{R}$ denotes the ReLU function defined as $\varphi(x) = \max\{x, 0\}$. The weight vector $\mathbf{w}_i$ (corresponding to hidden node $i$) and $\mathbf{P}(T_m)$ are defined as

$$\mathbf{w}_i = \begin{pmatrix} w_1^i \\ \vdots \\ w_d^i \end{pmatrix}, \qquad \mathbf{P}(T_m) = \begin{pmatrix} P(T_m, T_m + \delta_1) \\ \vdots \\ P(T_m, T_m + \delta_d) \end{pmatrix}$$

with $T_m < T_m + \delta_1 < \ldots < T_m + \delta_d \leq T_M$. Recall that, as a characteristic of the affine term structure model, the random variable $P(t, T)$ can be expressed as

$$P(t, T) = e^{A(t,T) - \sum_{i=1}^{d} B_i(t,T) x_i(t)}$$

for deterministic functions $A$ and $B_i$, which are available in closed form (see Brigo and Mercurio 2006). By the structure of the network, the weight vector is constrained to have only a single non-zero entry, which we will denote to have index $k$. Therefore, we can rewrite

$$H_i(t) = \mathbb{E}^{\mathbb{Q}} \left[ e^{-\int_t^{T_m} r(u)du} \max \left\{ w_i^k P(T_m, T_m + \delta_k) + b_i,\, 0 \right\} \Big| \mathcal{F}_t \right]$$

As we argued before, if $w_i^k$ and $b_i$ are both non-negative, $H_i(t)$ denotes the value of a forward contract. In that case, we have

$$
\begin{aligned}
H_i(t) &= \mathbb{E}^{\mathbb{Q}} \left[ e^{-\int_t^{T_m} r(u)du} \left( w_i^k P(T_m, T_m + \delta_k) + b_i \right) \Big| \mathcal{F}_t \right] \\
&= w_i^k \mathbb{E}^{\mathbb{Q}} \left[ e^{-\int_t^{T_m} r(u)du} \mathbb{E}^{\mathbb{Q}} \left[ e^{-\int_{T_m}^{T_m+\delta_k} r(u)du} \Big| \mathcal{F}_{T_m} \right] \Big| \mathcal{F}_t \right] + b_i \mathbb{E}^{\mathbb{Q}} \left[ e^{-\int_t^{T_m} r(u)du} \Big| \mathcal{F}_t \right] \\
&= w_i^k P(t, T_m + \delta_k) + b_i P(t, T_m)
\end{aligned}
$$

If, on the other hand, $b_i < 0 < w_i^k$ or $w_i^k < 0 < b_i$, we are dealing with a European call or put option, respectively. Closed-form expressions for European bond options are available based on Black's formula and have been treated extensively in the literature; see, for example, Musiela and Rutkowski (2005), Filipovic (2009), or Brigo and Mercurio (2006). In our case, we have

$$
H_i(t) = \begin{cases} w_i^k P(t, T_m + \delta_k) \Phi(d_+) + b_i P(t, T_m) \Phi(d_-) & \text{if } b_i < 0 < w_i^k \\ -b_i P(t, T_m) \Phi(-d_-) - w_i^k P(t, T_m + \delta_k) \Phi(-d_+) & \text{if } w_i^k < 0 < b_i \end{cases}
$$

where $\Phi$ denotes the CDF of a standard normal distribution, and we define

$$d_\pm := \frac{\log \left( -\frac{w_i^k P(t, T_m + \delta_k)}{b_i P(t, T_m)} \right) \pm \frac{1}{2} \Sigma(t, T_m)}{\sqrt{\Sigma(t, T_m)}}$$

and

$$\Sigma(t, T_m) := \int_t^{T_m} \| v(u, T_m + \delta_k) - v(u, T_m) \|^2 du$$

In the expression above, the function $v(t, T) \in \mathbb{R}^d$ refers to the instantaneous volatility at time $t$ of a discount bond maturing at $T$. Under the dynamics of Equation (A1), $v$ is given by

$$v(t, T) = \begin{pmatrix} \sum_{i=1}^{d} B_i(t, T) \sigma_{i1}(t) \\ \vdots \\ \sum_{i=1}^{d} B_i(t, T) \sigma_{id}(t) \end{pmatrix} \tag{A3}$$

*Appendix A.2. The Continuation Value with Fully Connected NN*

Once again, we consider the network $G_m(\cdot)$, focus on the outcome of hidden node $i \in \{1, \ldots, q\}$, and let $t \in [T_{m-1}, T_m)$. Now, our aim is to evaluate the conditional expectation below, which, by a change in numéraire argument, can be rewritten as

$$
\begin{aligned}
& \mathbb{E}^{\mathbb{Q}} \left[ e^{-\int_t^{T_m} r(u)du} \varphi(\mathbf{w}_i^\top \log \mathbf{P}(T_m) - b_i) \Big| \mathcal{F}_t \right] \\
& \quad = P(t, T_m) \mathbb{E}^{T_m} \left[ \max \left\{ \mathbf{w}_i^\top \log \mathbf{P}(T_m) - b_i),\, 0 \right\} \Big| \mathcal{F}_t \right]
\end{aligned}
$$

where the expectation on the right is taken under the $T_m$-forward measure, taking $P(t, T_m)$ as the numéraire. The weight vector $\mathbf{w}_i$ (corresponding to hidden node $i$) and $\log \mathbf{P}(T_m)$ are defined as

$$\mathbf{w}_i = \begin{pmatrix} w_1^i \\ \vdots \\ w_d^i \end{pmatrix}, \qquad \log \mathbf{P}(T_m) = \begin{pmatrix} \log P(T_m, T_m + \delta_1) \\ \vdots \\ \log P(T_m, T_m + \delta_d) \end{pmatrix}$$

with $T_m < T_m + \delta_1 < \ldots < T_m + \delta_d \leq T_M$. We set the input dimension equal to the number of risk factors (i.e., $d = n$). Therefore, we can write

$$\mathbf{w}_i^\top \log \mathbf{P}(T_m) = \sum_{j=1}^d w_j^i \log P(T_m, T_m + \delta_j)$$

$$= \sum_{j=1}^d w_j^i A(T_m, T_m + \delta_j) - \sum_{j=1}^d w_j^i \sum_{k=1}^d B_k(T_m, T_m + \delta_j) x_k(T_m)$$

$$= \begin{pmatrix} w_1^i & \cdots & w_d^i \end{pmatrix} \begin{pmatrix} A(T_m, T_m + \delta_1) \\ \vdots \\ A(T_m, T_m + \delta_d) \end{pmatrix}$$

$$- \begin{pmatrix} w_1^i & \cdots & w_d^i \end{pmatrix} \begin{pmatrix} B_1(T_m, T_m + \delta_1) & \cdots & B_d(T_m, T_m + \delta_1) \\ \vdots & \ddots & \vdots \\ B_1(T_m, T_m + \delta_d) & \cdots & B_d(T_m, T_m + \delta_d) \end{pmatrix} \begin{pmatrix} x_1(T_m) \\ \vdots \\ x_d(T_m) \end{pmatrix}$$

$$= \mathbf{w}_i^\top \mathbf{A}(T_m) - \mathbf{w}_i^\top \mathbf{B}(T_m) \mathbf{x}_{T_m}$$

where we implicitly define

$$\mathbf{A}(T_m) := \begin{pmatrix} A(T_m, T_m + \delta_1) \\ \vdots \\ A(T_m, T_m + \delta_d) \end{pmatrix},$$

$$\mathbf{B}(T_m) := \begin{pmatrix} B_1(T_m, T_m + \delta_1) & \cdots & B_d(T_m, T_m + \delta_1) \\ \vdots & \ddots & \vdots \\ B_1(T_m, T_m + \delta_d) & \cdots & B_d(T_m, T_m + \delta_d) \end{pmatrix}$$

In order to compute the conditional expectation of Equation (A4), a change in measure is required to obtain the dynamics of $x_1, \ldots, x_n$ under the $T_m-$forward measure. Consider the Radon–Nikodym derivative process Beyna (2013), defined by

$$\frac{d\mathbb{Q}^{T_m}}{d\mathbb{Q}}\bigg|\mathcal{F}_t = \frac{B(t)}{B(T_m)} \frac{P(T_m, T_m)}{P(t, T_m)} = \exp\left\{ -\int_t^{T_m} \nu(u, T_m) \cdot d\mathbf{W}(u) - \frac{1}{2}\int_t^{T_m} \|\nu(u, T_m)\|^2 du \right\}$$

where $\nu$ refers to to the instantaneous volatility of the numéraire, given in Equation (A3). The dynamics of the risk factors under $\mathbb{Q}^{T_m}$ can be obtained by an application of Girsanov's theorem Musiela and Rutkowski (2005). Denote by $\sigma_i(t) := (\sigma_{i1}(t), \ldots, \sigma_{id}(t))$ the $i$th row of the volatility matrix of $\mathbf{x}_t$ and let $\tilde{W}_i^{T_m}$ be Brownian motions under $\mathbb{Q}^{T_m}$; then,

$$dx_i(t) = -a_i(t)x_i(t)dt - \sigma_i(t) \cdot \nu(t, T_m)dt + \tilde{\sigma}_i(t)d\tilde{W}_i^{T_m}(t), \qquad i = 1, \ldots, d \tag{A4}$$

Let $\Theta_i(t, T_m) = \int_t^{T_m} \sigma_i(s) \cdot \nu(s, T_m) e^{-\int_s^{T_m} a_i(u)du} ds$; then, the SDE above solves to

$$x_i(T_m) = x_i(t)e^{-\int_t^{T_m} a_i(u)du} - \Theta_i(t, T_m) + \int_t^{T_m} \tilde{\sigma}_i(s)e^{-\int_s^{T_m} a_i(u)du} d\tilde{W}_i^{T_m}(s), \quad i = 1, \ldots, d \tag{A5}$$

It follows that, as a property of the Itô integral, the risk factors $(x_1(T_m), \ldots, x_n(T_m))$ as presented in Equation (A5), conditional on $\mathcal{F}_t$, have a multivariate normal distribution under $\mathbb{Q}^{T_m}$. Their mean vector and co-variance matrix are, respectively, given by

$$
\boldsymbol{\mu} := \begin{pmatrix} \mu_1 \\ \vdots \\ \mu_d \end{pmatrix} := \begin{pmatrix} \mathbb{E}^{T_m}[x_1(T_m)|\mathcal{F}_t] \\ \vdots \\ \mathbb{E}^{T_m}[x_d(T_m)|\mathcal{F}_t] \end{pmatrix} = \begin{pmatrix} x_1(t)e^{-\int_t^{T_m} a_1(u)du} - \Theta_1(t, T_m) \\ \vdots \\ x_d(t)e^{-\int_t^{T_m} a_d(u)du} - \Theta_d(t, T_m) \end{pmatrix}
$$

$$
\mathbf{C} := \begin{pmatrix} c_{11} & \cdots & c_{1d} \\ \vdots & \ddots & \vdots \\ c_{d1} & \cdots & c_{dd} \end{pmatrix} := \begin{pmatrix} \mathrm{Cov}[x_1(T_m), x_1(T_m)|\mathcal{F}_t] & \cdots & \mathrm{Cov}[x_1(T_m), x_d(T_m)|\mathcal{F}_t] \\ \vdots & \ddots & \vdots \\ \mathrm{Cov}[x_d(T_m), x_1(T_m)|\mathcal{F}_t] & \cdots & \mathrm{Cov}[x_d(T_m), x_d(T_m)|\mathcal{F}_t] \end{pmatrix}
$$

$$
c_{ii} = \int_t^{T_m} \tilde{\sigma}_i^2(s) e^{-2\int_s^{T_m} a_i(u)du} ds \qquad \forall_{i \in \{1, \ldots, d\}}
$$

$$
c_{ij} = \int_t^{T_m} \rho(s)\tilde{\sigma}_i(s)\tilde{\sigma}_j(s) e^{-\int_s^{T_m} (a_i(u)+a_j(u))du} ds \qquad \forall_{i \neq j}
$$

As a result, it should be clear that the random variable $Y := \mathbf{w}_i^\top \log \mathbf{P}(T_m)$ is normally distributed with mean and variance given, respectively, by

$$
\mu_Y = \mathbf{w}_i^\top \mathbf{A}(T_m) - \mathbf{w}_i^\top \mathbf{B}(T_m)\boldsymbol{\mu}
$$

and variance

$$
\sigma_Y^2 = \mathbf{w}_i^\top \mathbf{B}(T_m)\mathbf{C}\mathbf{B}(T_m)^\top \mathbf{w}_i
$$

As a result, we can compute

$$
\mathbb{E}^{\mathbb{Q}}\left[ e^{-\int_t^{T_m} r(u)du} \varphi(\mathbf{w}_i^\top \log \mathbf{P}(T_m) - b_i) \Big| \mathcal{F}_t \right] = P(t, T_m)\mathbb{E}^{T_m}\left[ \max(Y - b_i, 0) \big| \mathcal{F}_t \right]
$$

where the conditional expectation on the right-hand side can be expressed in closed form following a similar analysis as presented in Musiela and Rutkowski (2005). Let $d_i := \frac{\mu_Y - b_i}{\sigma_Y}$ and denote by $\xi \sim N(0, 1)$ a standard normal random variable. Then, it follows that

$$
\begin{aligned}
\mathbb{E}^{T_m}\left[ \max(Y - b_i, 0) \big| \mathcal{F}_t \right] &= \mathbb{E}^{T_m}\left[ (Y - b_i)\mathbb{1}_{\{Y > b_i\}} \big| \mathcal{F}_t \right] \\
&= \mathbb{E}^{T_m}\left[ (Y - \mu_Y)\mathbb{1}_{\{Y > b_i\}} \right] + (\mu_Y - b_i)\mathbb{Q}^{T_m}\left[ Y > b_i \big| \mathcal{F}_t \right] \\
&= \sigma_Y \mathbb{E}^{T_m}\left[ \frac{Y - \mu_Y}{\sigma_Y} \mathbb{1}_{\left\{ \frac{Y - \mu_Y}{\sigma_Y} > -d_i \right\}} \Big| \mathcal{F}_t \right] \\
&\quad + (\mu_Y - b_i)\mathbb{Q}^{T_m}\left[ \frac{Y - \mu_Y}{\sigma_Y} > -d_i \Big| \mathcal{F}_t \right] \\
&= \sigma_Y \mathbb{E}\left[ -\xi \mathbb{1}_{\{-\xi < d_i\}} \right] + (\mu_Y - b_i)\mathbb{P}[\xi < d_i] \\
&= \sigma_Y \phi(d_i) + (\mu_Y - b_i)\Phi(d_i)
\end{aligned}
$$

where $\phi$ denotes the standard normal density function and $\Phi$ the standard normal cumulative density function.

## Appendix B. Pre-Processing the Regression-Data

A procedure that significantly improves the fitting performance of the neural networks is the normalization of the training data. The linear rescaling of the input to the optimizer is a common form of data pre-processing Bishop et al. (1995). In the case of a multivariate input, the variables might have typical values in different orders of magnitude, even though that does not reflect their relative influence on determining the outcome Bishop et al. (1995). Normalizing the scale avoids the impact of a certain input being prioritized over another

input. Also, the transfer of the final weights in $G_{m+1}$ to the initialization of $G_m$ is more effective as the target variables are of roughly the same size at each time-step. In the default situation, the average continuation values would change in magnitude and the risk factor distribution would grow with each passing of a monitor date.

Another argument for pre-processing the input is that large data values typically induce large weights. Large weights can lead to exploding network outputs in the feed-forward process Goodfellow et al. (2016). Furthermore, it can cause an unstable optimization of the network, as extreme gradients can be very sensitive to small perturbations in the data Goodfellow et al. (2016).

In practice, we propose the following rescaling of the data. Denote by

$$
\hat{z}(T_m) := \left\{ \begin{pmatrix} z_1(T_m) \\ \vdots \\ z_d(T_m) \end{pmatrix}_1, \ldots, \begin{pmatrix} z_1(T_m) \\ \vdots \\ z_d(T_m) \end{pmatrix}_N \right\}, \quad \hat{V}(T_m) := \left\{ \tilde{V}\left(T_m; x_{T_m}^1\right), \ldots, \tilde{V}\left(T_m; x_{T_m}^N\right) \right\}
$$

the training points for the in- and output of network $G_m$. Define the standard sample mean and standard deviations as

$$
\mu_{z_i}(T_m) := \frac{1}{N} \sum_{n=1}^N z_i^n(T_m), \qquad \mu_V(T_m) := \frac{1}{N} \sum_{n=1}^N \tilde{V}\left(T_m; x_{T_m}^n\right)
$$

$$
\sigma_{z_i}(T_m) := \frac{1}{N-1} \sum_{n=1}^N (z_i^n(T_m) - \mu_{z_i})^2, \quad \sigma_V(T_m) := \frac{1}{N-1} \sum_{n=1}^N \left(\tilde{V}\left(T_m; x_{T_m}^n\right) - \mu_V(T_m)\right)^2
$$

We then perform a simple element-wise linear transformation to obtain the scaled data $\hat{z}^\dagger$ and $\hat{V}^\dagger$ given by

$$
\hat{z}_i^\dagger(T_m) := \frac{\hat{z}_i(T_m) - \mu_{z_i}(T_m)}{\sigma_{z_i}(T_m)}, \qquad \hat{V}^\dagger(T_m) := \frac{\hat{V}(T_m)}{\sigma_V(T_m)}
$$

With the transformations above in mind, it is important to adjust the associated composition of the replicating portfolio accordingly. For the two network designs, this has the following implications:

**The locally connected NN case:** Consider the outcome of the $i^{th}$ hidden node $\nu_i$ and denote the input of the network as $\mathbf{z}$. Then, $\nu_i = \varphi\left(w_i^k z_k + b_i\right)$, where $k$ is the index of the only non-zero entry of $\mathbf{w}_i$, the $i^{th}$ row of weight matrix $\mathbf{w}_1$. The transformation $\mathbf{z} \mapsto \frac{\mathbf{z} - \mu_{\mathbf{z}}}{\sigma_{\mathbf{z}}}$ implies that

$$
\nu_i \mapsto \varphi\left(w_i^k \frac{z_k - \mu_{z_k}}{\sigma_{z_k}} + b_i\right) = \varphi\left(\frac{w_i^k}{\sigma_{z_k}} z_k + \left(b_i - \frac{w_i^k \mu_{z_k}}{\sigma_{z_k}}\right)\right)
$$

As a consequence, in the analysis of Appendix A.1, the transformations $w_i^k \mapsto \frac{w_i^k}{\sigma_{z_k}}$ and $b_i \mapsto b_i - \frac{w_i^k \mu_{z_k}}{\sigma_{z_k}}$ should be taken into account. Additionally, the transformation $\mathbf{w}_2 \mapsto \sigma_V \mathbf{w}_2$ is required to account for the scaling of $\hat{V}$.

**The fully connected NN case:** Again, consider the outcome of the $i^{th}$ hidden node $\nu_i$. This time, the transformation $\mathbf{z} \mapsto \frac{\mathbf{z} - \mu_{\mathbf{z}}}{\sigma_{\mathbf{z}}}$ implies that

$$
\nu_i \mapsto \varphi\left(\mathbf{w}_i^\top \frac{\mathbf{z} - \mu_{\mathbf{z}}}{\sigma_{\mathbf{z}}} + b_i\right) = \varphi\left(\sum_{j=1}^d \frac{w_i^j}{\sigma_{z_j}} z_j + \left(b_i - \sum_{j=1}^d \frac{w_i^j \mu_{z_j}}{\sigma_{z_i}}\right)\right)
$$

As a consequence, in the analysis of Appendix A.2, the transformations $\mathbf{w}_i \mapsto \left( \frac{w_i^1}{\sigma_{z_1}}, \dots \right.$

$\left. , \frac{w_i^d}{\sigma_{z_d}} \right)^\top$ and $b_i \mapsto b_i - \sum_{j=1}^d \frac{w_i^j \mu_{z_j}}{\sigma_{z_i}}$ should be taken into account. And, again, the transformation $\mathbf{w}_2 \mapsto \sigma_V \mathbf{w}_2$ is required to account for the scaling of $\hat{V}$.

## Appendix C. Hyperparameter Selection

The accuracy of the neural network fitting procedure is dependent on the choice of several hyperparameters. For the numerical experiments reported in Section 6, the hyperparameters have been selected based on a convergence analysis. We focused on the following:

- Hidden node count: see Figure A1;
- Size training set: see Figure A2;
- Learning-rate: see Figure A3.

Several numerical experiments indicated that the batch size did not have a significant impact on the fitting accuracy and is therefore fixed at a default of 32. For the convergence analysis of the parameters listed above, we considered a $1Y \times 10Y$ receiver Bermudan swaption with a fixed rate of $K = 0.03$. Experiments were performed under the two-factor G2++ model using the model specifications depicted in Table 3. The figures show the mean absolute errors of the neural network fits per monitor date in basis points of the notional.

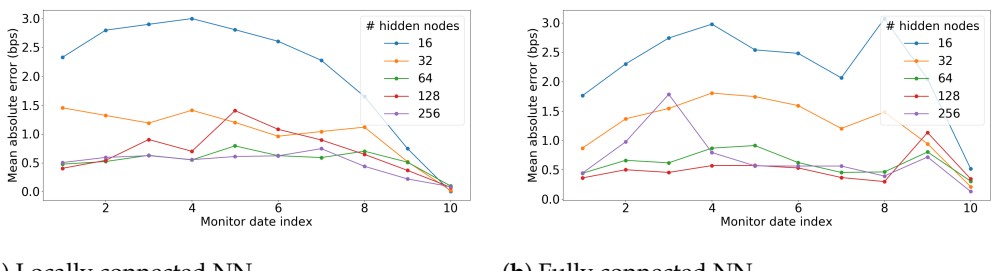

(**a**) Locally connected NN        (**b**) Fully connected NN

**Figure A1.** Impact hidden node count: accuracy of the neural network fit per monitor date under a 2-factor model. # training points = 5000. Learning-rate = 0.0002.

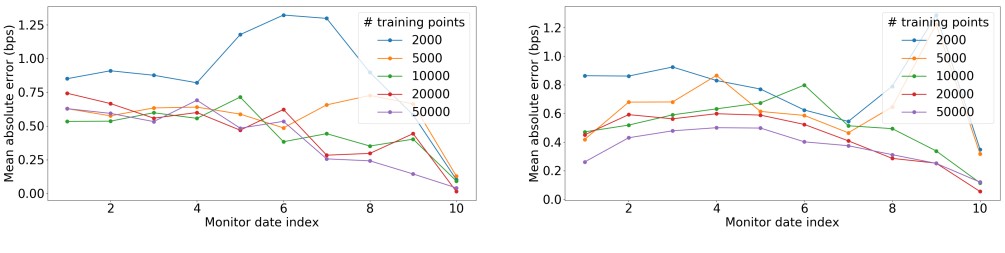

(**a**) Locally connected NN        (**b**) Fully connected NN

**Figure A2.** Impact size training set: accuracy of the neural network fit per monitor date under a 2-factor model. # hidden nodes = 64. Learning-rate = 0.0002.

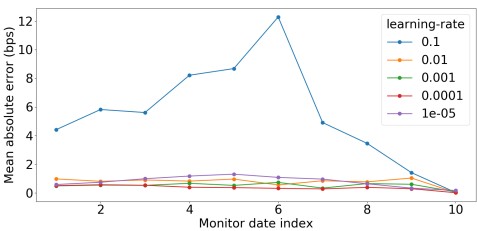
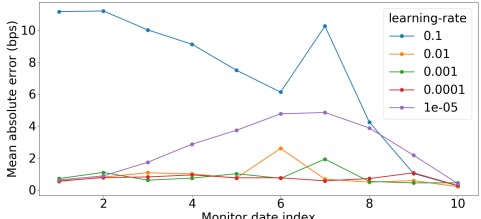

(**a**) Locally connected NN          (**b**) Fully connected NN

**Figure A3.** Impact learning-rate: accuracy of the neural network fit per monitor date under a 2-factor model. # hidden nodes = 64. # training-points = 10,000.

## Appendix D. Proof of Theorem 1

**Proof.** We prove by induction on $m$. At the last exercise date of the Bermudan, i.e., $t = T_{M-1}$, we have $V(T_{M-1}; x) = \tilde{V}(T_{M-1}; x) := \max\{h_{M-1}(x), 0\}$, representing the final pay-off of the contract, which at $T_{M-1}$ is exactly known. Hence, it should be obvious that

$$\sup_{x \in \mathcal{I}_d} B^{-1}(T_{M-1}) \big| V(T_{M-1}; x) - \tilde{V}(T_{M-1}; x) \big| = 0$$

For the inductive step, assume that, for some $T_{m+1} \in \mathcal{T}_f$, an approximation $\tilde{V}(T_{m+1})$ of the price is given, satisfying

$$\sup_{x \in \mathcal{I}_d} B^{-1}(T_{m+1}) \big| V(T_{m+1}; x) - \tilde{V}(T_{m+1}; x) \big| < k\varepsilon$$

We will show that it follows that, for all $t \in [T_m, T_{m+1})$,

$$\sup_{x \in \mathcal{I}_d} B^{-1}(t) \big| V(t; x) - \tilde{V}(t; x) \big| < (k+1)\varepsilon$$

First, consider the case $t \in (T_m, T_{m+1})$. It follows that

$$
\begin{aligned}
\sup_{x \in \mathcal{I}_d} \left| \frac{V(t; x) - \tilde{V}(t; x)}{B(t)} \right| &= \sup_{x \in \mathcal{I}_d} \left| \frac{C_m(t; x) - \tilde{C}_m(t; x)}{B(t)} \right| \\
&= \sup_{x \in \mathcal{I}_d} \left| \mathbb{E}^{\mathbb{Q}} \left[ \frac{V(T_{m+1})}{B(T_{m+1})} \bigg| \mathbf{x}_t = x \right] - \mathbb{E}^{\mathbb{Q}} \left[ \frac{G_{m+1}(z_{m+1})}{B(T_{m+1})} \bigg| \mathbf{x}_t = x \right] \right| \\
&\leq \sup_{x \in \mathcal{I}_d} \mathbb{E}^{\mathbb{Q}} \left[ B^{-1}(T_{m+1}) |V(T_{m+1}) - G_{m+1}(z_{m+1})| \bigg| \mathbf{x}_t = x \right] \\
&= \sup_{x \in \mathcal{I}_d} \mathbb{E}^{\mathbb{Q}} \Big[ B^{-1}(T_{m+1}) \big| V(T_{m+1}) - \tilde{V}(T_{m+1}) \\
&\qquad\qquad + \tilde{V}(T_{m+1}) - G_{m+1}(z_{m+1}) \big| \bigg| \mathbf{x}_t = x \Big] \\
&\leq \sup_{x \in \mathcal{I}_d} \bigg( \mathbb{E}^{\mathbb{Q}} \left[ B^{-1}(T_{m+1}) \big| V(T_{m+1}) - \tilde{V}(T_{m+1}) \big| \bigg| \mathbf{x}_t = x \right] \\
&\qquad\qquad + \mathbb{E}^{\mathbb{Q}} \left[ B^{-1}(T_{m+1}) \big| \tilde{V}(T_{m+1}) - G_{m+1}(z_{m+1}) \big| \bigg| \mathbf{x}_t = x \right] \bigg)
\end{aligned}
$$

In the last expression above, the first term is bounded due to the induction hypothesis, i.e., $B^{-1}(T_{m+1}) \big| V(T_{m+1}) - \tilde{V}(T_{m+1}) \big| < k\varepsilon$. The second term is bounded by assumption, i.e., there exists a network $G_{m+1}(\cdot)$ such that $B^{-1}(T_{m+1}) \big| \tilde{V}(T_{m+1}) - G_{m+1}(z_{m+1}) \big| < \varepsilon$. We hence conclude that

$$\sup_{x \in \mathcal{I}_d} B^{-1}(t) \big| V(t; x) - \tilde{V}(t; x) \big| < (k+1)\varepsilon, \qquad \forall_{t \in (T_m, T_{m+1})}$$

If, on the other hand, $t = T_m$, we have that

$$\sup_{x \in \mathcal{I}_d} \left| \frac{V(t;x) - \tilde{V}(t;x)}{B(t)} \right| = \sup_{x \in \mathcal{I}_d} \left| \frac{\max\{C_m(t;x), h_m(x)\} - \max\{\tilde{C}_m(t;x), h_m(x)\}}{B(t)} \right|$$

Denoting $H(x) := B^{-1}(t) \left| \max\{C_m(t;x), h_m(x)\} - \max\{\tilde{C}_m(t;x), h_m(x)\} \right|$ in the expression above, we can distinguish four cases for each $x \in \mathcal{I}_d$, which are

- $C_m(t;x), \tilde{C}_m(t;x) > h_m(x)$, then $H(x) = B^{-1}(t) \left| C_m(t;x) - \tilde{C}_m(t;x) \right| < (k+1)\varepsilon$;
- $C_m(t;x), \tilde{C}_m(t;x) < h_m(x)$, then $H(x) = B^{-1}(t) |h_m(x) - h_m(x)| = 0 < (k+1)\varepsilon$;
- $C_m(t;x) < h_m(x) < \tilde{C}_m(t;x)$, then $H(x) = B^{-1}(t) \left| h_m(x) - \tilde{C}_m(t;x) \right|$
  $< B^{-1}(t) \left| C_m(t;x) - \tilde{C}_m(t;x) \right| < (k+1)\varepsilon$;
- $\tilde{C}_m(t;x) < h_m(x) < C_m(t;x)$, then $H(x) = B^{-1}(t) |C_m(t;x) - h_m(x)|$
  $< B^{-1}(t) \left| C_m(t;x) - \tilde{C}_m(t;x) \right| < (k+1)\varepsilon$.

From all the cases, we can induce that

$$\sup_{x \in \mathcal{I}_d} B^{-1}(t) \left| V(t;x) - \tilde{V}(t;x) \right| \leq (k+1)\varepsilon$$

We conclude that, by induction on $m = M - 1, \dots, 0$,

$$\sup_{x \in \mathcal{I}_d} B^{-1}(t) \left| V(t;x) - \tilde{V}(t;x) \right| < M\varepsilon$$

for all $t \in [0, T_{M-1}]$. $\square$

**Appendix E. Proof of Theorem 2**

**Proof.** First, we fix some notation.

- Let $V_m := V(T_m)$ denote the true price of the Bermudan swaption at $T_m$ conditioned on the fact that it is not yet exercised.
- Let $\tilde{C}_m := B(T_m) \mathbb{E}^{\mathbb{Q}} \left[ \frac{G_{m+1}(z_{m+1})}{B(T_{m+1})} \middle| \mathcal{F}_{T_m} \right]$ denote the estimator of the continuation value at $T_m$.
- Let $\tilde{V}_m := \max\{\tilde{C}_m, h_m(\mathbf{x}_{T_m})\}$ denote the estimator of $V_m$.
- Let $G_m := G_m(z_m)$ denote the neural network approximation of $\tilde{V}_m$.
- Let $B_m := B(T_m)$ denote the numéraire at $T_m$.
- Let $h_m := h_m(\mathbf{x}_{T_m})$.

Let $T_m \in \{T_0, \dots, T_{M-1}\}$. We will prove the theorem by induction on $m$. For the base case, note that at time zero we have

$$\left| V(0) - \tilde{V}(0) \right| = \left| \mathbb{E}^{\mathbb{Q}} \left[ \frac{V_0}{B_0} \middle| \mathcal{F}_0 \right] - \mathbb{E}^{\mathbb{Q}} \left[ \frac{G_0}{B_0} \middle| \mathcal{F}_0 \right] \right| \leq \mathbb{E}^{\mathbb{Q}} \left[ \left| \frac{V_0 - G_0}{B_0} \right| \middle| \mathcal{F}_0 \right] \tag{A6}$$

which is induced by Jensen's inequality. For the inductive step, assume that, for some $m \in \{0, \dots, M-1\}$, we have that

$$\left| V(0) - \tilde{V}(0) \right| < \mathbb{E}^{\mathbb{Q}} \left[ \left| \frac{V_m - G_m}{B_m} \right| \middle| \mathcal{F}_0 \right] + m \cdot \varepsilon \tag{A7}$$

The expectation in (A7) can be rewritten using the triangular inequality

$$\mathbb{E}^{\mathbb{Q}} \left[ \left| \frac{V_m - G_m}{B_m} \right| \middle| \mathcal{F}_0 \right] = \mathbb{E}^{\mathbb{Q}} \left[ \left| \frac{V_m - \tilde{V}_m + \tilde{V}_m - G_m}{B_m} \right| \middle| \mathcal{F}_0 \right]$$
$$\leq \mathbb{E}^{\mathbb{Q}} \left[ \left| \frac{V_m - \tilde{V}_m}{B_m} \right| \middle| \mathcal{F}_0 \right] + \mathbb{E}^{\mathbb{Q}} \left[ \left| \frac{\tilde{V}_m - G_m}{B_m} \right| \middle| \mathcal{F}_0 \right] \tag{A8}$$

The second term in (A8) is, by assumption, bounded by $\varepsilon$. Note that the first term in (A8) can be bounded as

$$\mathbb{E}^{\mathbb{Q}}\left[\left.\left|\frac{V_m - \tilde{V}_m}{B_m}\right|\right|\mathcal{F}_0\right] = \mathbb{E}^{\mathbb{Q}}\left[\left.\left|\frac{\max\{C_m, h_m\} - \max\{\tilde{C}_m, h_m\}}{B_m}\right|\right|\mathcal{F}_0\right]$$

$$\leq \mathbb{E}^{\mathbb{Q}}\left[\left.\left|\frac{C_m - \tilde{C}_m}{B_m}\right|\right|\mathcal{F}_0\right]$$

$$= \mathbb{E}^{\mathbb{Q}}\left[\left.\left|\mathbb{E}^{\mathbb{Q}}\left[\left.\frac{V_{m+1}}{B_{m+1}}\right|\mathcal{F}_{T_m}\right] - \mathbb{E}^{\mathbb{Q}}\left[\left.\frac{G_{m+1}}{B_{m+1}}\right|\mathcal{F}_{T_m}\right]\right|\right|\mathcal{F}_0\right]$$

$$\leq \mathbb{E}^{\mathbb{Q}}\left[\left.\mathbb{E}^{\mathbb{Q}}\left[\left.\left|\frac{V_{m+1} - G_{m+1}}{B_{m+1}}\right|\right|\mathcal{F}_{T_m}\right]\right|\mathcal{F}_0\right]$$

$$= \mathbb{E}^{\mathbb{Q}}\left[\left.\left|\frac{V_{m+1} - G_{m+1}}{B_{m+1}}\right|\right|\mathcal{F}_0\right]$$

It follows that

$$\left|V(0) - \tilde{V}(0)\right| < \mathbb{E}^{\mathbb{Q}}\left[\left.\left|\frac{V_{m+1} - G_{m+1}}{B_{m+1}}\right|\right|\mathcal{F}_0\right] + (m+1)\cdot\varepsilon$$

For the final step, note that if $m = M - 1$, we have

$$\mathbb{E}^{\mathbb{Q}}\left[\left.\left|\frac{V_m - G_m}{B_m}\right|\right|\mathcal{F}_0\right] = \mathbb{E}^{\mathbb{Q}}\left[\left.\left|\frac{\max\{h_{M-1}, 0\} - G_{M-1}}{B_{M-1}}\right|\right|\mathcal{F}_0\right] < \varepsilon$$

We conclude by induction on $m$ that $\left|V(0) - \tilde{V}(0)\right| < M\varepsilon$ $\quad\square$

**Appendix F. Proof of Theorem 3**

**Proof.** We consider the following three events: $\{\tau = \tilde{\tau}\}$, $\{\tau < \tilde{\tau}\}$, and $\{\tau > \tilde{\tau}\}$. Note that

$$V(0) - L(0) = \mathbb{E}^{\mathbb{Q}}\left[\left.\frac{h_\tau(\mathbf{x}_\tau)}{B(\tau)} - \frac{h_{\tilde{\tau}}(\mathbf{x}_{\tilde{\tau}})}{B(\tilde{\tau})}\right|\mathcal{F}_0\right]$$

$$= \mathbb{E}^{\mathbb{Q}}\left[\left.\left(\frac{h_\tau(\mathbf{x}_\tau)}{B(\tau)} - \frac{h_{\tilde{\tau}}(\mathbf{x}_{\tilde{\tau}})}{B(\tilde{\tau})}\right)\mathbb{1}_{\{\tau=\tilde{\tau}\}}\right|\mathcal{F}_0\right] + \mathbb{E}^{\mathbb{Q}}\left[\left.\left(\frac{h_\tau(\mathbf{x}_\tau)}{B(\tau)} - \frac{h_{\tilde{\tau}}(\mathbf{x}_{\tilde{\tau}})}{B(\tilde{\tau})}\right)\mathbb{1}_{\{\tau<\tilde{\tau}\}}\right|\mathcal{F}_0\right]$$

$$+ \mathbb{E}^{\mathbb{Q}}\left[\left.\left(\frac{h_\tau(\mathbf{x}_\tau)}{B(\tau)} - \frac{h_{\tilde{\tau}}(\mathbf{x}_{\tilde{\tau}})}{B(\tilde{\tau})}\right)\mathbb{1}_{\{\tau>\tilde{\tau}\}}\right|\mathcal{F}_0\right]$$

$$= E_1 + E_2 + E_3$$

We will bound the three terms above one by one.
Bounding $\mathbf{E_1}$: Starting with the event $\{\tau = \tilde{\tau}\}$, we observe that we can write

$$E_1 = \mathbb{E}^{\mathbb{Q}}\left[\left.\left(\frac{h_\tau(\mathbf{x}_\tau)}{B(\tau)} - \frac{h_\tau(\mathbf{x}_\tau)}{B(\tau)}\right)\mathbb{1}_{\{\tau=\tilde{\tau}\}}\right|\mathcal{F}_0\right] = 0$$

Bounding $\mathbf{E_2}$: We continue with the event $\{\tau < \tilde{\tau}\}$. For this, we will introduce two types of sub-events: $A_m := \{\tau = T_m \wedge \tilde{\tau} > T_m\}$ and $B_m := \{\tau \leq T_m \wedge \tilde{\tau} > T_m\}$, where $\wedge$ denotes the logical AND operator. Also, we define the difference process $e_m := \frac{\tilde{V}(T_m)}{B(T_m)} - \frac{h_{\tilde{\tau}}(\mathbf{x}_{\tilde{\tau}})}{B(\tilde{\tau})}$. It should be clear that $\mathbb{1}_{\{\tau<\tilde{\tau}\}} = \sum_{m=0}^{M-1}\mathbb{1}_{A_m}$. Therefore, it holds that

$$E_2 = \sum_{m=0}^{M-1}\mathbb{E}^{\mathbb{Q}}\left[\left.\left(\frac{h_\tau(\mathbf{x}_\tau)}{B(\tau)} - \frac{h_{\tilde{\tau}}(\mathbf{x}_{\tilde{\tau}})}{B(\tilde{\tau})}\right)\mathbb{1}_{A_m}\right|\mathcal{F}_0\right] \leq \sum_{m=0}^{M-1}\mathbb{E}^{\mathbb{Q}}\left[\left.e_m\mathbb{1}_{A_m}\right|\mathcal{F}_0\right]$$

where the inequality follows from the fact that the direct estimator has the property $\tilde{V}(T_m) = \max\{\tilde{C}_m, h_m\} \geq h_m$. Now, we will show by induction that $E_2 < (M-1)\varepsilon$. First, observe that $A_0 \equiv B_0$. Second, note that, for any $m \in \{0, \ldots, M-1\}$, we have that

$$
\begin{aligned}
\mathbb{E}^{\mathbb{Q}}\left[e_m \mathbb{1}_{B_m} \Big| \mathcal{F}_0\right] &= \mathbb{E}^{\mathbb{Q}}\left[\left(\mathbb{E}^{\mathbb{Q}}\left[\frac{G_{m+1}(z_{m+1})}{B(T_{m+1})} \Big| \mathcal{F}_{T_m}\right] - \frac{h_{\tilde{\tau}}(\mathbf{x}_{\tilde{\tau}})}{B(\tilde{\tau})}\right) \mathbb{1}_{B_m} \Big| \mathcal{F}_0\right] \\
&= \mathbb{E}^{\mathbb{Q}}\left[\left(\frac{G_{m+1}(z_{m+1})}{B(T_{m+1})} - \frac{h_{\tilde{\tau}}(\mathbf{x}_{\tilde{\tau}})}{B(\tilde{\tau})}\right) \mathbb{1}_{B_m} \Big| \mathcal{F}_0\right] \\
&\leq \mathbb{E}^{\mathbb{Q}}\left[\left|\frac{G_{m+1}(z_{m+1})}{B(T_{m+1})} - \frac{\tilde{V}(T_{m+1})}{B(T_{m+1})}\right| \mathbb{1}_{B_m} \Big| \mathcal{F}_0\right] + \mathbb{E}^{\mathbb{Q}}\left[e_{m+1} \mathbb{1}_{B_m} \Big| \mathcal{F}_0\right]
\end{aligned}
\tag{A9}
$$

The first equality follows from the fact that $\tilde{V}(T_m) = \tilde{C}_m$ in the event $\tilde{\tau} > T_m$. The second equality follows from the tower rule in combination with the fact that $\mathbb{1}_{B_m}$ is $\mathcal{F}_{T_m}$−measurable. The final inequality follows from an application of the triangle inequality. The first term in (A9) is, by assumption, bounded by $\varepsilon$. The second term in (A9) can be rewritten by observing that $\mathbb{1}_{B_m} := \mathbb{1}_{B_m^1} + \mathbb{1}_{B_m^2} := \mathbb{1}_{\{\tau \leq T_m \wedge \tilde{\tau} = T_{m+1}\}} + \mathbb{1}_{\{\tau \leq T_m \wedge \tilde{\tau} > T_{m+1}\}}$. We have that

$$
\mathbb{E}^{\mathbb{Q}}\left[e_{m+1} \mathbb{1}_{B_m^1} \Big| \mathcal{F}_0\right] = \mathbb{E}^{\mathbb{Q}}\left[\left(\frac{h_{m+1}(\mathbf{x}_{T_{m+1}})}{B(T_{m+1})} - \frac{h_{m+1}(\mathbf{x}_{T_{m+1}})}{B(T_{m+1})}\right) \mathbb{1}_{B_m^1} \Big| \mathcal{F}_0\right] = 0
$$

Furthermore, we have that $\mathbb{1}_{B_m^2} + \mathbb{1}_{A_{m+1}} = \mathbb{1}_{B_{m+1}}$. Therefore we can infer that

$$
\begin{aligned}
\mathbb{E}^{\mathbb{Q}}\left[e_m \mathbb{1}_{B_m} \Big| \mathcal{F}_0\right] + \mathbb{E}^{\mathbb{Q}}\left[e_{m+1} \mathbb{1}_{A_{m+1}} \Big| \mathcal{F}_0\right] &< \varepsilon + \mathbb{E}^{\mathbb{Q}}\left[e_{m+1} \mathbb{1}_{B_m^2} \Big| \mathcal{F}_0\right] + \mathbb{E}^{\mathbb{Q}}\left[e_{m+1} \mathbb{1}_{A_{m+1}} \Big| \mathcal{F}_0\right] \\
&= \varepsilon + \mathbb{E}^{\mathbb{Q}}\left[e_{m+1} \mathbb{1}_{B_{m+1}} \Big| \mathcal{F}_0\right]
\end{aligned}
$$

Together with the fact that $A_0 \equiv B_0$, we conclude by induction on $m$ that

$$
\begin{aligned}
E_2 &\leq \mathbb{E}^{\mathbb{Q}}\left[e_0 \mathbb{1}_{B_0} \Big| \mathcal{F}_0\right] + \sum_{m=1}^{M-1} \mathbb{E}^{\mathbb{Q}}\left[e_m \mathbb{1}_{A_m} \Big| \mathcal{F}_0\right] \\
&< \varepsilon + \mathbb{E}^{\mathbb{Q}}\left[e_1 \mathbb{1}_{B_1} \Big| \mathcal{F}_0\right] + \sum_{m=2}^{M-1} \mathbb{E}^{\mathbb{Q}}\left[e_m \mathbb{1}_{A_m} \Big| \mathcal{F}_0\right] \\
&\quad\vdots \\
&< (M-1)\varepsilon + \mathbb{E}^{\mathbb{Q}}\left[e_{M-1} \mathbb{1}_{B_{M-1}} \Big| \mathcal{F}_0\right] = (M-1)\varepsilon
\end{aligned}
$$

Bounding **E₃**: We finalize the proof by considering the third event $\{\tau > \tilde{\tau}\}$. In a similar fashion as before, we introduce two types of sub-events: $A_m := \{\tilde{\tau} = T_m \wedge \tau > T_m\}$ and $B_m := \{\tilde{\tau} \leq T_m \wedge \tau > T_m\}$. Also, again define a difference process, this time given by $e_m := \frac{h_\tau(\mathbf{x}_\tau)}{B(\tau)} - \frac{\tilde{V}(T_m)}{B(T_m)}$. It should be clear that $\mathbb{1}_{\{\tau > \tilde{\tau}\}} = \sum_{m=0}^{M-1} \mathbb{1}_{A_m}$. Therefore, it holds that

$$
E_3 = \sum_{m=0}^{M-1} \mathbb{E}^{\mathbb{Q}}\left[\left(\frac{h_\tau(\mathbf{x}_\tau)}{B(\tau)} - \frac{h_{\tilde{\tau}}(\mathbf{x}_{\tilde{\tau}})}{B(\tilde{\tau})}\right) \mathbb{1}_{A_m} \Big| \mathcal{F}_0\right] = \sum_{m=0}^{M-1} \mathbb{E}^{\mathbb{Q}}\left[e_m \mathbb{1}_{A_m} \Big| \mathcal{F}_0\right]
$$

where the second equality follows from the fact that the direct estimator has the property $\tilde{V}(\tilde{\tau}) = h_{\tilde{\tau}}$. Now, we will show by induction that $E_3 < (M-1)\varepsilon$. Note that, for any $m \in \{0, \ldots, M-1\}$, we have that

$$
\begin{aligned}
\mathbb{E}^{\mathbb{Q}}\!\left[e_m \mathbb{1}_{B_m} \Big| \mathcal{F}_0\right] &\leq \mathbb{E}^{\mathbb{Q}}\!\left[\left(\frac{h_\tau(\mathbf{x}_\tau)}{B(\tau)} - \mathbb{E}^{\mathbb{Q}}\!\left[\frac{G_{m+1}(z_{m+1})}{B(T_{m+1})}\Big| \mathcal{F}_{T_m}\right]\right)\mathbb{1}_{B_m} \Big| \mathcal{F}_0\right] \\
&= \mathbb{E}^{\mathbb{Q}}\!\left[\left(\frac{h_\tau(\mathbf{x}_\tau)}{B(\tau)} - \frac{G_{m+1}(z_{m+1})}{B(T_{m+1})}\right)\mathbb{1}_{B_m} \Big| \mathcal{F}_0\right] \\
&\leq \mathbb{E}^{\mathbb{Q}}\!\left[\left|\frac{\tilde{V}(T_{m+1})}{B(T_{m+1})} - \frac{G_{m+1}(z_{m+1})}{B(T_{m+1})}\right|\mathbb{1}_{B_m} \Big| \mathcal{F}_0\right] + \mathbb{E}^{\mathbb{Q}}\!\left[e_{m+1} \mathbb{1}_{B_m} \Big| \mathcal{F}_0\right]
\end{aligned}
\tag{A10}
$$

The first inequality follows from the fact that $\tilde{V}(T_m) = \max\{\tilde{C}_m, h_m\} \geq \tilde{C}_m$. The subsequent equality follows from the tower rule in combination with the fact that $\mathbb{1}_{B_m}$ is $\mathcal{F}_{T_m}-$measurable. The final inequality follows from an application of the triangle inequality. The first term in (A10) is, by assumption, bounded by $\varepsilon$. The second term in (A10) can be rewritten by observing that $\mathbb{1}_{B_m} := \mathbb{1}_{B_m^1} + \mathbb{1}_{B_m^2} := \mathbb{1}_{\{\tilde{\tau} \leq T_m \wedge \tau = T_{m+1}\}} + \mathbb{1}_{\{\tilde{\tau} \leq T_m \wedge \tau > T_{m+1}\}}$. We have that

$$
\mathbb{E}^{\mathbb{Q}}\!\left[e_{m+1} \mathbb{1}_{B_m^1} \Big| \mathcal{F}_0\right] = \mathbb{E}^{\mathbb{Q}}\!\left[\left(\frac{h_{m+1}(\mathbf{x}_{T_{m+1}})}{B(T_{m+1})} - \frac{\tilde{V}(T_{m+1})}{B(T_{m+1})}\right)\mathbb{1}_{B_m^1} \Big| \mathcal{F}_0\right] \leq 0
$$

where the inequality follows from the fact that $\tilde{V}(T_{m+1}) = \max\{\tilde{C}_{m+1}, h_{m+1}\} \geq h_{m+1}$. Furthermore, we have that $\mathbb{1}_{B_m^2} + \mathbb{1}_{A_{m+1}} = \mathbb{1}_{B_{m+1}}$. Therefore, we can once again infer that

$$
\begin{aligned}
\mathbb{E}^{\mathbb{Q}}\!\left[e_m \mathbb{1}_{B_m} \Big| \mathcal{F}_0\right] + \mathbb{E}^{\mathbb{Q}}\!\left[e_{m+1} \mathbb{1}_{A_{m+1}} \Big| \mathcal{F}_0\right] &< \varepsilon + \mathbb{E}^{\mathbb{Q}}\!\left[e_{m+1} \mathbb{1}_{B_m^2} \Big| \mathcal{F}_0\right] + \mathbb{E}^{\mathbb{Q}}\!\left[e_{m+1} \mathbb{1}_{A_{m+1}} \Big| \mathcal{F}_0\right] \\
&= \varepsilon + \mathbb{E}^{\mathbb{Q}}\!\left[e_{m+1} \mathbb{1}_{B_{m+1}} \Big| \mathcal{F}_0\right]
\end{aligned}
$$

Together with the fact that $A_0 \equiv B_0$, we again conclude by induction on $m$ that

$$
\begin{aligned}
E_3 &\leq \mathbb{E}^{\mathbb{Q}}\!\left[e_0 \mathbb{1}_{B_0} \Big| \mathcal{F}_0\right] + \sum_{m=1}^{M-1} \mathbb{E}^{\mathbb{Q}}\!\left[e_m \mathbb{1}_{A_m} \Big| \mathcal{F}_0\right] \\
&< \varepsilon + \mathbb{E}^{\mathbb{Q}}\!\left[e_1 \mathbb{1}_{B_1} \Big| \mathcal{F}_0\right] + \sum_{m=2}^{M-1} \mathbb{E}^{\mathbb{Q}}\!\left[e_m \mathbb{1}_{A_m} \Big| \mathcal{F}_0\right] \\
&\vdots \\
&< (M-1)\varepsilon + \mathbb{E}^{\mathbb{Q}}\!\left[e_{M-1} \mathbb{1}_{B_{M-1}} \Big| \mathcal{F}_0\right] = (M-1)\varepsilon
\end{aligned}
$$

Conclusion: We hence find that

$$
V(0) - L(0) = E_1 + E_2 + E_3 < 0 + (M-1)\varepsilon + (M-1)\varepsilon = 2(M-1)\varepsilon
$$

$\square$

### Appendix G. Proof of Theorem 4

**Proof.** The discounted true price process is a supermartingale under $\mathbb{Q}$. Therefore, we have that $\frac{V(t)}{B(t)} = Y_t + Z_t$ for a martingale $Y_t$ and a predictable process $Z_t$, which starts at zero (i.e., $Z_0 = 0$) and is strictly decreasing. Define a difference process on $\mathcal{T}$, given

by $e_{T_m} = \frac{V(T_m) - G_m(z_m)}{B(T_m)}$. We can rewrite martingale $M_t$ as defined in (13) in terms of $e_t$ as follows:

$$M_{T_m} = \frac{G_0(z_0)}{B(T_0)} + \sum_{j=1}^{m} \left( \frac{G_j(z_j)}{B(T_j)} - \mathbb{E}^{\mathbb{Q}} \left[ \frac{G_j(z_j)}{B(T_j)} \bigg| \mathcal{F}_{T_{j-1}} \right] \right)$$

$$= Y_{T_m} - e_{T_0} - \sum_{j=1}^{m} \left( e_{T_j} - \mathbb{E}^{\mathbb{Q}} \left[ e_{T_j} | \mathcal{F}_{T_{j-1}} \right] \right)$$

Substituting the expression for $M_t$ into the definition of $U(0)$ yields

$$U(0) = M_0 + \mathbb{E}^{\mathbb{Q}} \left[ \max_{T_m \in \mathcal{T}_f} \left\{ \frac{h_m(\mathbf{x}_{T_m})}{B(T_m)} - M_{T_m} \right\} \bigg| \mathcal{F}_0 \right]$$

$$= \mathbb{E}^{\mathbb{Q}} \left[ \frac{G_0(z_0)}{B(T_0)} \bigg| \mathcal{F}_0 \right] + \mathbb{E}^{\mathbb{Q}} \left[ \max_{m \in \{0,\dots,M-1\}} \left\{ \frac{h_m(\mathbf{x}_{T_m})}{B(T_m)} - Y_{T_m} + e_{T_0} \right. \right.$$

$$\left. \left. + \sum_{j=1}^{m} \left( e_{T_j} - \mathbb{E}^{\mathbb{Q}} \left[ e_{T_j} | \mathcal{F}_{T_{j-1}} \right] \right) \right\} \bigg| \mathcal{F}_0 \right]$$

$$\leq \mathbb{E}^{\mathbb{Q}} \left[ \frac{V(T_0)}{B(T_0)} \bigg| \mathcal{F}_0 \right] + \mathbb{E}^{\mathbb{Q}} \left[ \max_{m \in \{0,\dots,M-1\}} \left\{ \sum_{j=1}^{m} \left( e_{T_j} - \mathbb{E}^{\mathbb{Q}} \left[ e_{T_j} | \mathcal{F}_{T_{j-1}} \right] \right) \right\} \bigg| \mathcal{F}_0 \right]$$

The last step follows by merging $\mathbb{E}^{\mathbb{Q}} \left[ e_{T_0} | \mathcal{F}_0 \right]$ with $M_0$ and by noting that $\frac{h_m(\mathbf{x}_{T_m})}{B(T_m)} - Y_{T_m} \leq \frac{V(T_m)}{B(T_m)} - Y_{T_m} = Z_{T_m} \leq 0$. The remaining inequality is not easy to bound Andersen and Broadie (2004). However, by taking the absolute values of the difference process, we can obtain a loose bound as follows:

$$U(0) \leq V(0) + \mathbb{E}^{\mathbb{Q}} \left[ \max_{m \in \{0,\dots,M-1\}} \left\{ \sum_{j=1}^{m} \left| e_{T_j} \right| + \sum_{j=1}^{m} \left| \mathbb{E}^{\mathbb{Q}} \left[ e_{T_j} | \mathcal{F}_{T_{j-1}} \right] \right| \right\} \bigg| \mathcal{F}_0 \right]$$

$$\leq V(0) + \mathbb{E}^{\mathbb{Q}} \left[ \sum_{j=1}^{M-1} \left| e_{T_j} \right| + \sum_{j=1}^{M-1} \left| \mathbb{E}^{\mathbb{Q}} \left[ e_{T_j} | \mathcal{F}_{T_{j-1}} \right] \right| \bigg| \mathcal{F}_0 \right]$$

$$\leq V(0) + 2 \sum_{j=1}^{M-1} \mathbb{E}^{\mathbb{Q}} \left[ \left| e_{T_j} \right| \big| \mathcal{F}_0 \right]$$

Note that, as a consequence of Theorem 2, we have that $\mathbb{E}^{\mathbb{Q}} \left[ \left| e_{T_m} \right| \big| \mathcal{F}_0 \right] < (M-m)\varepsilon$. It follows that

$$|U(0) - V(0)| < 2 \sum_{m=1}^{M-1} (M-m)\varepsilon = M(M-1)\varepsilon$$

This concludes the proof. $\square$

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
