# Peer review of "A Semi-Static Replication Method for Bermudan Swaptions under an Affine Multi-Factor Model"

_risks, doi:10.3390/risks11100168_

Round 1

Reviewer 1 Report

Please, see the file attached.

Reviewer 2 Report

This paper provides a semi-static replication algorithm for Bermudan swaptions under an affine, multi-factor term-structure model. The paper is interesting. 

My main comment is on the error analysis. In the section on error analysis, the error estimates are given in terms of M and epsilon, however M and epsilon are correlated. I think when epsilon is small, M needs to be big. The authors need to discuss this. 

The paper is well written in general, but a careful proof reading will be help to improve the presentation. For example, 

line 5 of the Introduction, It is better to change "Most of these metrics depend on the distributions of the potential future ..." to "Most of these metrics depend on the distribution of the potential future ..."

line 8 of the Introduction, change "The latter is generally considered the involved aspect," to " he latter is generally considered the most involved aspect,"?

Round 2

Reviewer 1 Report

The paper has been greatly improved. Let me suggest the acceptance of it.